# The role of the interactome in the maintenance of deleterious variability in human populations

Luz Garcia-Alonso[1], Jorge Jiménez-Almazán[1,2], Jose Carbonell-Caballero[1], Alicia Vela-Boza[3], Javier Santoyo-López[3], Guillermo Antiñolo[3,4,5] & Joaquin Dopazo[1,2,3,6,*]

## Abstract

Recent genomic projects have revealed the existence of an unexpectedly large amount of deleterious variability in the human genome. Several hypotheses have been proposed to explain such an apparently high mutational load. However, the mechanisms by which deleterious mutations in some genes cause a pathological effect but are apparently innocuous in other genes remain largely unknown. This study searched for deleterious variants in the 1,000 genomes populations, as well as in a newly sequenced population of 252 healthy Spanish individuals. In addition, variants causative of monogenic diseases and somatic variants from 41 chronic lymphocytic leukaemia patients were analysed. The deleterious variants found were analysed in the context of the interactome to understand the role of network topology in the maintenance of the observed mutational load. Our results suggest that one of the mechanisms whereby the effect of these deleterious variants on the phenotype is suppressed could be related to the configuration of the protein interaction network. Most of the deleterious variants observed in healthy individuals are concentrated in peripheral regions of the interactome, in combinations that preserve their connectivity, and have a marginal effect on interactome integrity. On the contrary, likely pathogenic cancer somatic deleterious variants tend to occur in internal regions of the interactome, often with associated structural consequences. Finally, variants causative of monogenic diseases seem to occupy an intermediate position. Our observations suggest that the real pathological potential of a variant might be more a systems property rather than an intrinsic property of individual proteins.

**Keywords** exome sequencing; interactome; mutational load; network analysis; robustness
**Subject Categories** Network Biology; Chromatin, Epigenetics, Genomics & Functional Genomics

**Mol Syst Biol.** (2014) 10: 752

## Introduction

The outcome of several international collaborative projects published recently (Durbin *et al*, 2010; Dunham *et al*, 2012; Fu *et al*, 2013) has revealed the existence of an enormous amount of variation at genome level in apparently normal, healthy individuals. A specific type of variant, known as loss-of-function (LoF), which is thought to severely affect the function of human protein-coding genes (MacArthur & Tyler-Smith, 2010), is of particular importance. Traditionally, these variants have been associated with severe Mendelian diseases due to their potentially deleterious effect. Actually, the existence of this mutational load has been known for a long time, and different estimations of its magnitude have been made, ranging from < 10 genes carrying deleterious mutations (Muller, 1950) to almost one hundred (Kondrashov, 1995). However, recent observations from several genome sequencing projects (Durbin *et al*, 2010; Hudson *et al*, 2010; Dunham *et al*, 2012) which report an unexpectedly large number of these variants in the genomes of apparently healthy individuals seem to contradict this view. Conservative estimations suggest that there are no < 250 LoF variants per sequenced genome, 100 of them known to be related to human diseases, and more than 30 in a homozygous state and predicted to be highly damaging (Xue *et al*, 2012), suggesting a previously unnoticed level of variation with putative functional consequences in protein-coding regions in humans (MacArthur & Tyler-Smith, 2010). Moreover, this apparently pathological variation is not restricted to coding regions but also seems to occur in other non-coding, regulatory elements, such as miRNAs (Carbonell *et al*, 2012), transcription factor binding sites (TFBSs) (Spivakov *et al*, 2012) and others (Lappalainen *et al*, 2013). The origin of this

1 Computational Genomics Department, Centro de Investigación Príncipe Felipe (CIPF), Valencia, Spain
2 Bioinformatics of Rare Diseases (BIER), CIBER de Enfermedades Raras (CIBERER), Valencia, Spain
3 Medical Genome Project, Genomics and Bioinformatics Platform of Andalusia (GBPA), Seville, Spain
4 Department of Genetics, Reproduction and Fetal Medicine, Institute of Biomedicine of Seville, University Hospital Virgen del Rocío/Consejo Superior de Investigaciones Científicas/University of Seville, Seville, Spain
5 Centro de Investigación Biomédica en Red de Enfermedades Raras (CIBERER), Seville, Spain
6 Functional Genomics Node, (INB) at CIPF, Valencia, Spain
*Corresponding author. Tel: +34 96 328 9680; E-mail: jdopazo@cipf.es

apparent excess of LoF variants has been attributed to the combination of a recent accelerated human population growth with a weak purifying selection (Keinan & Clark, 2012; Tennessen *et al*, 2012). Different reasons could account for the maintenance of such a large number of LoF variants in apparently healthy individuals, including severe recessive disease alleles in homozygous state; late onset phenotypes; reduced penetrance phenotypes which require additional genetic and/or environmental factors for expression; gene redundancy and even sequencing errors (Nothnagel *et al*, 2011; MacArthur *et al*, 2012; Xue *et al*, 2012). Although the possibility that any of the donors of these genome projects eventually become ill cannot be ruled out, these were non-vulnerable adults and it seems unlikely that they have suffered extensively from any genetic disease (Xue *et al*, 2012). The paradox of apparently healthy individuals carrying an excess of deleterious mutations has led to the recategorization of known disease-causing mutations (Xue *et al*, 2012) and the reconsideration of the putative functional effect of some apparently deleterious variants (Nothnagel *et al*, 2011). However, the mechanisms by which specific deleterious variants can have a clear pathological effect when affecting some genes while in others they are apparently innocuous remain largely unknown (Nothnagel *et al*, 2011; Xue *et al*, 2012).

The notion of cell functionality as a consequence of the complex interactions between their molecular components is not new (Hartwell *et al*, 1999) and was proposed more than a decade ago in the context of systems biology (Kitano, 2002). These interacting components define operational entities or modules to which different elementary functions can be attributed. In practical terms, the network of protein–protein interactions, or interactome, has been used extensively as a theoretical scaffold interrelating proteins. The interactome has been used to define sub-networks of interacting proteins associated with features in genomic experiments (Ideker & Sharan, 2008), which can be considered to be functional modules (Dittrich *et al*, 2008). Connections within the interactome are organized so as to make the system robust and preserve stable phenotypes under changing conditions and attacks. Several studies have demonstrated that biological networks are topologically robust against the removal of a certain number of nodes. Thus is due to the fact that even though a few nodes concentrate most of the network edges, the great majority are slightly connected (known as Power-Law networks) (Albert *et al*, 2000; Jeong *et al*, 2001). However, this robustness has limits and the higher the number of removed nodes is, the more vulnerable the network becomes (Agoston *et al*, 2005). A study of double synthetic lethality in yeast revealed that in some cases, individual removal of any of the two nodes has no effect on the yeast metabolic network, while combined removal produced a severe disruption in the information flow (Segre *et al*, 2005; Costanzo *et al*, 2010). Further observations in yeast (Fraser & Plotkin, 2007; McGary *et al*, 2007) and other model organisms such as worm (Lee *et al*, 2008) also support the idea that genes producing a similar mutant phenotype are tightly linked in the interactome.

Under this perspective, diseases can be understood as disruptions of functional modules, supporting the idea of a modular nature of human genetic diseases (Oti & Brunner, 2007; Oti *et al*, 2008). This modularity, extensively described in numerous reports (Brunner & van Driel, 2004; Gandhi *et al*, 2006; Lim *et al*, 2006; Goh *et al*, 2007; Wagner *et al*, 2007; Ideker & Sharan, 2008; Mitra *et al*, 2013), suggests that causative genes for the same disease often reside in the same biological module, which can be a protein complex (Lage *et al*, 2007), or a subnetwork of protein interactions (Lim *et al*, 2006; Ideker & Sharan, 2008; Vidal *et al*, 2011). It has also been described that disease genes tend to be connected to other disease genes (Goh *et al*, 2007; Lage *et al*, 2007; Wagner *et al*, 2007), which tend to be co-expressed and display coherent functions according to GO annotations (Ideker & Sharan, 2008; Montaner *et al*, 2009). Similarly, cancer gene products are highly connected and are centrally located in the network (Jonsson & Bates, 2006; Hernandez *et al*, 2007; Rambaldi *et al*, 2008). As a consequence of the robustness of the modules, the vast majority of disease phenotypes are rarely caused by the failure of a unique gene product, but rather reflect various pathological processes which interact in a complex network (Barabasi *et al*, 2011). An obvious example is cancer, where a succession of mutations is necessary for a cell to acquire oncogenic potential (Hanahan & Weinberg, 2011).

From an evolutionary point of view, it has been observed that proteins under positive selection tend to be located at the periphery of the global protein interaction network, while central network proteins, which are likely to have a larger portion of their surface involved in interactions, tend to be under negative selection (Fraser *et al*, 2002; Kim *et al*, 2007). In general, central proteins carry out more essential functions (e.g. tumour suppressors or oncogenes), while peripheral proteins tend to be non-essential disease genes (D'Antonio & Ciccarelli, 2011; Serra *et al*, 2011; Vidal *et al*, 2011).

Given the role of the interactome in assuring the robustness of cell systems against mutations, our hypothesis is that the actual interactome topology could be buffering the impact of deleterious variants, thus permitting what seems to be a high mutation load. In order to check the extent to which this hypothesis is compatible with recent observations on human variability (MacArthur & Tyler-Smith, 2010; MacArthur *et al*, 2012; Xue *et al*, 2012), the coding sequences (exomes) of 1,330 healthy individuals were analysed to study the impact of the actual levels of variability on interactome properties. The sequences included samples from thirteen worldwide distributed populations from the 1,000 genomes project (Durbin *et al*, 2010) as well as whole exome sequencing (WES) data corresponding to 252 healthy Spanish samples from the Medical Genome Project (http://www.medical genomeproject.com). The results were compared to paired WES of matched tumour and normal samples from 41 individuals with chronic lymphocytic leukaemia (Quesada *et al*, 2012), from the International Cancer Genome Consortium. The analysis yielded findings that allow explaining the existence of a genetic load of this magnitude. For example, proteins carrying deleterious variants in healthy individuals tend to have fewer connections than unaffected proteins, especially when such variants affect the protein in homozygosis. However, the most interesting observation is that most of the apparent deleterious mutational load observed in healthy individuals tends to occur in peripheral regions of the interactome, preserving its integrity. On the contrary, mutations with pathological consequences are more frequently observed in proteins located in internal regions of the interactome.

# Results

## Deleterious variants in proteins of the interactome observed in the populations

The model of the human interactome used here was built using data on protein–protein interactions from the BioGRID (Chatr-Aryamontri *et al*, 2013), IntAct (Kerrien *et al*, 2012) and MINT (Licata *et al*, 2012) databases. To avoid possible false positives or experimental errors, only interactions detected by at least two different detection methods (von Mering *et al*, 2002) were used. The resulting curated interactome consists of a total of 7,331 proteins connected by 21,623 interactions (see Materials and Methods). Figure 1 shows the average number of variants per individual in the proteins that define the interactome used in this study. As it has been previously described for the complete set of human proteins in several reports of genomic variability, African populations show higher variability (over 8,000 variants) than the rest of the populations (about 6,500 variants), including the CLL genomes (Fig 1A).

Among these, there are variants with a clear deleterious effect, such as stop loss, stop gain and splicing disrupting conserved variants. In addition, any non-synonymous variant with a SIFT score lower than 0.05 or a Polyphen score higher than 0.95 was considered as deleterious, as recommended in the original publications (Ramensky *et al*, 2002; Kumar *et al*, 2009). Since the application of both scores sometimes results in contradictory predictions (Hicks *et al*, 2011), an *in silico* study was performed on a subset of 20 randomly chosen variants (eight predicted to be non-damaging, five somatic predicted as damaging from CLL and seven predicted as damaging from non-disease populations). Table 1 shows the relationship between the predictions derived from SIFT and Polyphen and the structural features calculated for the subset of selected variants. In general, a good agreement between predicted deleterious effect and unfavourable changes in the sequence and structure properties can be observed. Figure 2 depicts an example of this agreement. The average number of potentially deleterious variants (Fig 1B) follows a similar pattern to the total number of variants (Fig 1A). African populations undergo more mutational load compared to the rest of the populations. The same pattern is observed for the number of proteins affected by deleterious variants in heterozygous state (Fig 1C). As expected, the Spanish population sequenced here presented a level of variation similar to that observed in non-African populations. However, this pattern is inverted when proteins with deleterious variants in homozygosis are analysed (Fig 1D). This observation is compatible with the history of the populations, with an older African population which has cumulated more variability but has filtered out deleterious variants in homozygosis, whereas the rest of the populations underwent a relatively recent bottleneck which is reflected in a lower level of variability and a higher level of homozygosity (Lohmueller *et al*, 2008). This genetic fingerprint is still observable in the proteins of the interactome.

Supplementary Table S1 describes 34,220 deleterious variants found in the sequenced Spanish population, which can also be found on a web server (http://spv.babelomics.org/) in which variation can be queried in an interactive manner.

## Proteins affected by deleterious variants in normal populations, monogenic diseases and cancer patients have different topological roles

We analysed the occurrence of deleterious mutations in proteins with different network properties in the interactome. Figure 3A shows the number of interactions corresponding to the proteins affected by a deleterious variant either in both alleles (homozygosis) or in only one allele (heterozygosis) or not affected by any deleterious variant, in at least one individual. It also shows the number of interactions observed in proteins with deleterious somatic mutations in CLL, proteins corresponding to monogenic diseases (see Supplementary Table S2) and the subset of somatic mutations in CLL corresponding to cancer driver genes (Vogelstein *et al*, 2013) (see Supplementary Table S3). The number of interactions in proteins with both alleles affected by a deleterious variant in healthy individuals was significantly lower than the number of interactions observed either in proteins with only one allele affected (FDR-adjusted Mann–Whitney *U*-test *P*-value = 0.000544) or in unaffected proteins (*P*-value = $5.22 \times 10^{-5}$). Proteins carrying only one allele affected by a deleterious variant showed a slightly lower number of interactions than unaffected proteins, although the difference is not significant in this case, probably because they have no pathogenic effect in either case. In a scenario of mutational disease represented by all the CLL proteins carrying somatic mutations (driver and passenger variants), the number of interactions in affected proteins was significantly higher than in healthy homozygote (*P*-value = $1.49 \times 10^{-5}$) and the healthy heterozygote (*P*-value = 0.00169) scenarios, as expected. The proteins affected by monogenic diseases displayed a significantly higher number of connections than the CLL proteins carrying somatic mutations (*P*-value = 0.0265) (and obviously more than the deleterious homozygous and heterozygous and unaffected proteins in healthy individuals, see Fig 3B). However, if only cancer driver proteins carrying somatic deleterious mutations in CLL are considered, the number of connections was significantly higher than any other subset of proteins analysed, including monogenic disease proteins (see Fig 3B). The analysis of the relationship between the same sets of genes and other properties such as betweenness (Fig 3C and D) and closeness centrality (Fig 3E and F) was repeated, obtaining a similar trend. The results demonstrate a clear relationship between the degree of pathogenicity of the scenario and the connectivity of the proteins affected.

Figure 4 depicts how the number of connections, the closeness centrality and the betweenness present a weak, but significant negative correlation (Spearman's rank correlation coefficient $\rho = -0.0661$, *P*-value = $1.34 \times 10^{-7}$, $\rho = -0.0536$, *P*-value = $1.934 \times 10^{-5}$ and $\rho = -0.0534$, *P*-value = $2.053 \times 10^{-5}$, respectively) with the frequency of occurrence of deleterious variants in the population (both in homozygosis and in heterozygosis). This trend, although negative as well, is not significant in the case of homozygosis, probably due to the lower sample size. On the contrary, in the pathological condition represented by CLL, the network properties number of connections ($\rho = 0.152$, *P*-value = 0.0116), betweenness ($\rho = 0,118$, *P*-value = 0.051) and closeness centrality ($\rho = 0.128$, *P*-value = 0.0335) are positively correlated with the recurrence of the mutation across patients.

Previous evolutionary studies documented a preferential occurrence of adaptive events at the periphery of the human protein

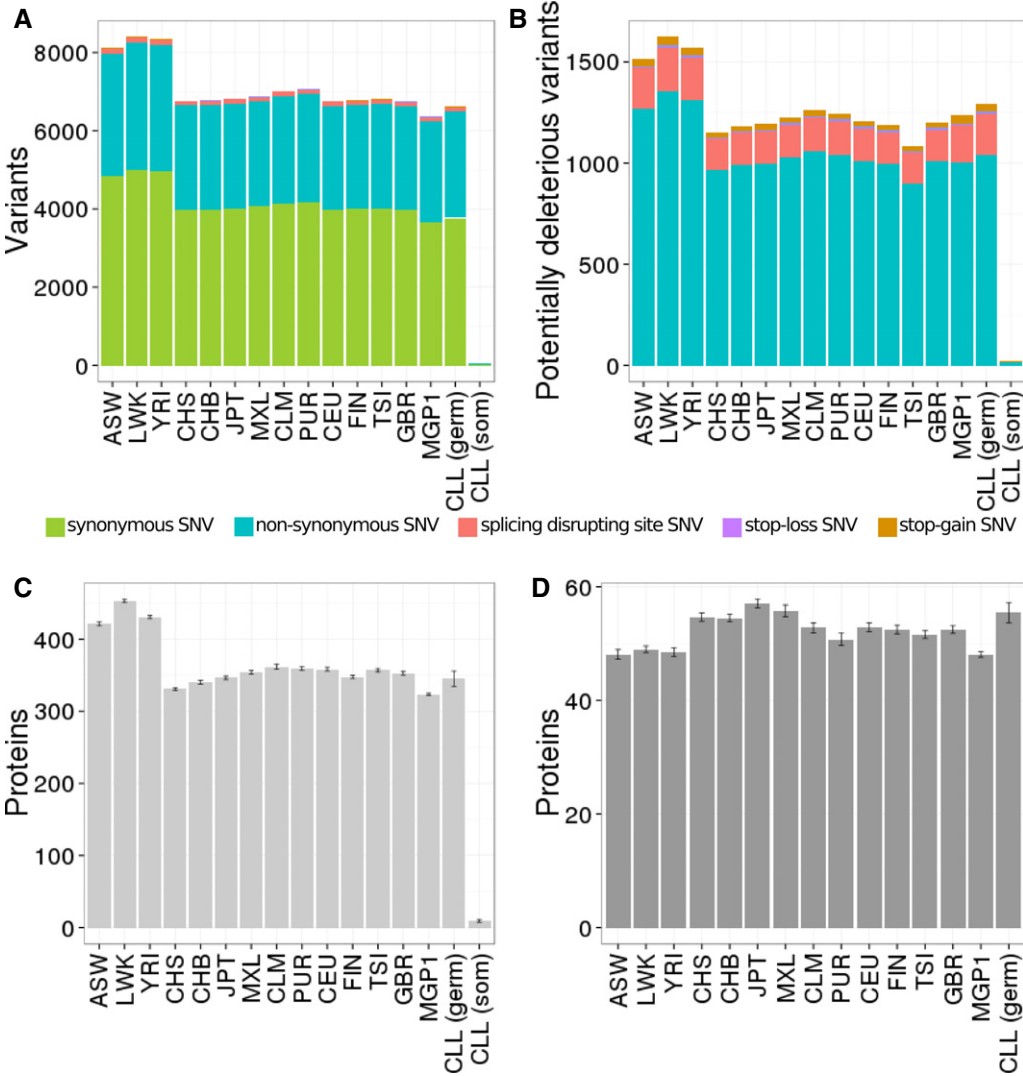

**Figure 1.  Summary of variants found in the proteins which configure the human interactome in all the populations analysed.**

A   Number of variants found.
B   Number of potentially deleterious variants.
C   Number of proteins carrying at least one deleterious variant in one of their alleles (mutation load).
D   Number of proteins carrying deleterious variants in both alleles (homozygous mutation load).

Data information: Bars represent the population average value and errors represent the dispersion found in the different individual sequences analysed.

interaction network (Fraser *et al*, 2002; Kim *et al*, 2007). It was confirmed that the distribution of selective pressures, measured as the ratio of non-synonymous to synonymous variants, across the network properties used here (number of interactions, betweenness and closeness centrality) was consistent with what was previously observed: proteins under positive selection tend to be placed in the periphery of the network, while proteins under negative selection tend to be in the internal regions (see Supplementary Fig S1).

**Effect of deleterious variants observed in normal individuals on the interactome structure**

The effect that the specific combination of deleterious variants carried by any healthy individual has on the interactome was

examined. Since loss-of-function variants were considered, the recessive (and most plausible) scenario was tested. This was achieved by removing proteins from the interactome when they were affected by deleterious variants in both alleles (homozygous for the alternative allele). Then, the impact that this subtraction had on the interactome structure was calculated (see Materials and Methods, section 'Selection of deleterious variants' for details). The impact is inferred by measuring the changes in several global network properties such as the number of connections, the average length of shortest paths and the number of components. These parameters account for the interconnectedness and integrity of the interactome (Albert *et al*, 2000). The values obtained for these parameters in the 1,000 genomes and MGP1 populations correspond to interactomes of healthy individuals.

**Table 1. Relationship between the predictions derived from SIFT and Polyphen and the structural features calculated for the selected subset of variants.**

Unfavourable structural properties of the mutations are highlighted in bold and underlined, and partially unfavourable properties are highlighted in bold. The remaining properties were neutral. The first column contains the mutation; the second column contains the gene name; the third column contains the PDB identifier of the structure template; the fourth column lists the SIFT scores; the fifth column lists the Polyphen scores; the sixth column contains the prediction according to the SIFT and Polyphen scores as: ND: non-damaging and D: deleterious; the seventh column contains the polarity change in a hydrophobicity scale (see Materials and Methods); the eighth column contains the change in the protein charge, based on changes in the change of the changed residue (see Materials and Methods); the ninth column lists the SNAP prediction, N: neutral and NN: non-neutral, with the accuracy in brackets; the tenth column contains the SMD prediction N: neutral, S: stabilizing, SS: slightly stabilizing, D: destabilizing, SD: slightly destabilizing and HD: highly stabilizing and causing protein malfunction; the eleventh column contains the SMD pseudo ΔΔG; and the twelfth column lists the change in percentage of solvent accessibility, coded as A: accessible, PA: partly accessible and B: buried.

| Mutation | Gene | PDB ID | SIFT | PolyPhen | Prediction | Polarity | Charge | SNAP | SMD | | | Secondary structure |
| | | | | | | | | | PRED | ΔΔG | % Solvent accessibility | |
| --- | --- | --- | --- | --- | --- | --- | --- | --- | --- | --- | --- | --- |
| T503A | SPG7 | 2QZ4_A | 0.37 | 0.001 | ND | 1/1 | 0/0 | N (53%) | N | 0.39 | 63.8 A/55.9 PA | h-bonded turn |
| R171Q | NFRKB | 3u21_A | 0.36 | 0.2249 | ND | 2/2 | **+/0** | **NN (58%)** | N | 0.23 | 101.5 A/111.1 A | Bend |
| K267R | CA4 | 1ZNC_A | NA | 0.314 | ND | 2/2 | +/+ | N (92%) | **S** | **1.2** | 26.8 PA/33.8 PA | Loop or irregular |
| A7S | PDIA3 | 3F8U_A | 0.1 | 0 | ND | 1/1 | 0/0 | N (92%) | SD | −0.56 | 110.5 A/103.6 A | Loop or irregular |
| K215R | RBBP4 | 2XU7_A | 0.28 | 0 | ND | 2/2 | +/+ | N (85%) | SD | 0.92 | 41.4 PA/57.3 PA | Extended strand |
| L56M | RET | 2X2U_A | 0.46 | 0.003 | ND | 0/0 | 0/0 | N (89%) | SD | −0.69 | 3.6 B/4 B | Extended strand |
| R48K | LXN | 2BO9_D | 0.95 | 0 | ND | 2/2 | +/+ | N (92%) | SD | −0.64 | 53.7 PA/66.5 PA | Loop or irregular |
| P459L | BACE2 | 3ZKM_A | 0.62 | 0.397 | ND | **1/0** | 0/0 | N (69%) | SS | 0.81 | 33.8 PA/48 PA | Loop or irregular |
| Y623C | SF3B1 | 2FHO_A | 0 | 0.999 | **D** | 0/0 | 0/0 | **NN (68%)** | **HD** | **2.21** | 33.6 PA/45.5 PA | Alpha helix |
| T663I | SF3B1 | 2FHO_A | 0 | 0.998 | **D** | **1/0** | 0/0 | **NN (58%)** | **HD** | **2.58** | 6.8 B/8.6 B | Alpha helix |
| K700E | SF3B1 | 2FHO_A | 0 | 0.999 | **D** | 2/2 | **+/−** | **NN (58%)** | SS | 0.6 | 47 PA/38.2 PA | Alpha helix |
| E372G | PABPC1 | 4F02_A | 0 | 0.86 | **D** | **2/1** | **−/0** | **NN (82%)** | **HD** | **−2.8** | **72.2 A/58 PA** | Alpha helix |
| R374C | PABPC1 | 4F02_A | 0 | 0.998 | **D** | **2/0** | **+/0** | **NN (78%)** | N | −0.2 | **59.3 A/52 PA** | Alpha helix |
| E114G | SDCBP | 1N99_A | 0 | 0.234 | **D** | **2/1** | **−/0** | N (53%) | **D** | **−1.79** | 62.8 A/72.4 A | Extended strand |
| R251G | RP2 | 3BH6_B | 0.05 | 0.998 | **D** | 2/1 | **+/0** | **NN (58%)** | **HD** | **−3.25** | 29.4 PA/66.9 PA | Alpha helix |
| T207M | AGT | 2WXW_A | NA | 0.991 | **D** | **1/0** | 0/0 | **NN (82%)** | **S** | **1.41** | 5.2 B/2.6 B | Extended strand |
| E322D | ATF4 | 1Cl6_A | 0.01 | 0.998 | **D** | 2/2 | −/− | N (78%) | **D** | **−1.75** | **77.4 PA/86.1 A** | Alpha helix |
| T287M | GRB7 | 4K81_A | 0.03 | 0.895 | **D** | **1/0** | 0/0 | **NN (70%)** | **S** | **1.35** | 47.9 PA/55 PA | Loop or irregular |
| C79F | MAVS | 3J6C_A | 0 | 0.999 | **D** | 0/0 | 0/0 | **NN (82%)** | **HD** | **−2.03** | 10.8 B/8.8 B | Loop or irregular |
| T124I | BAIAP2L1 | 2KXC_A | 0.05 | 0.042 | **D** | **1/0** | 0/0 | N (53%) | **S** | **1.94** | 44.9 PA/59.5 PA | Alpha helix |

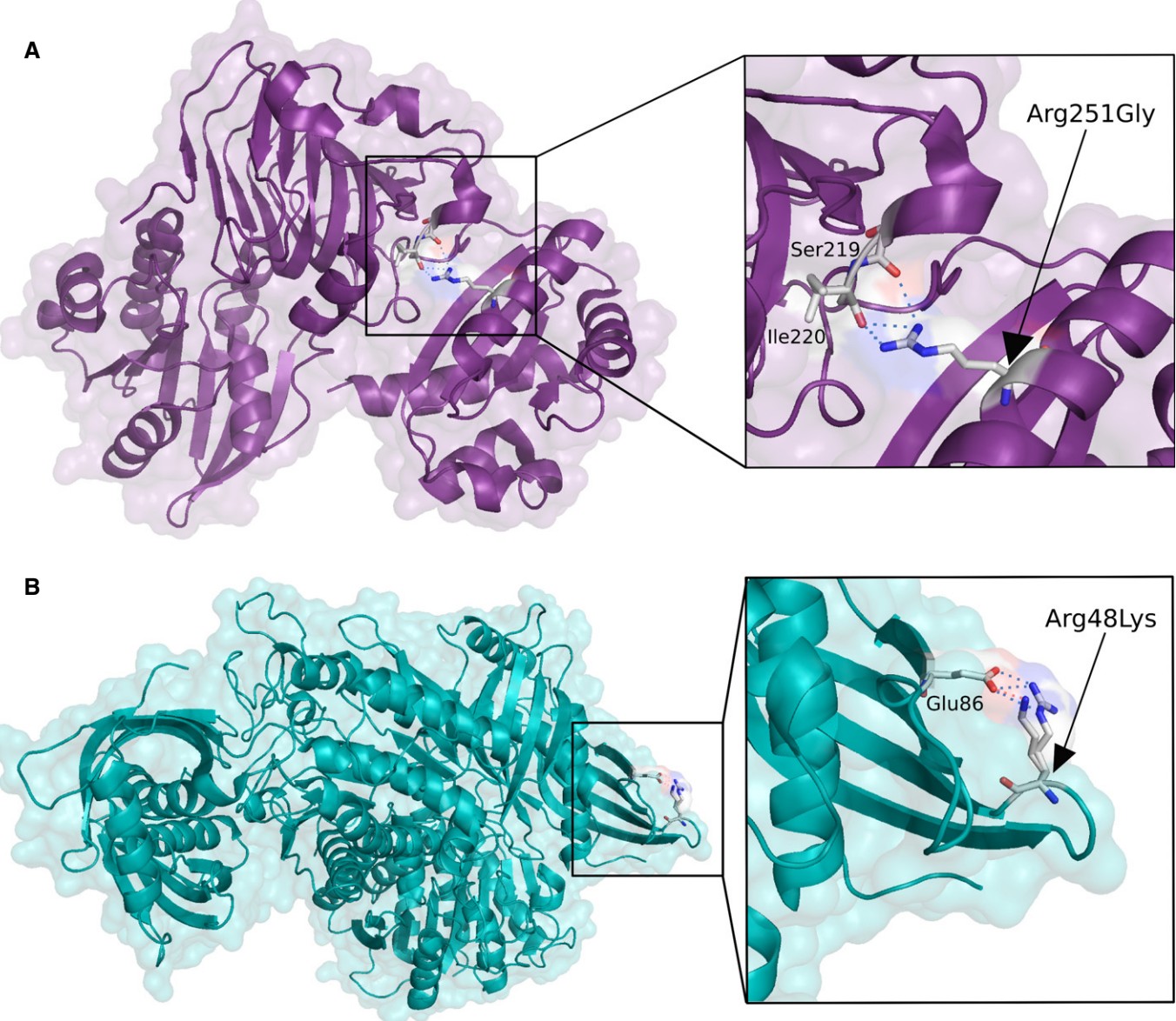

**Figure 2.  Molecular model of the human RP2 (A) and LXN (B) proteins and detailed view of the altered amino acids (Arg251Gly and Arg48Lys, respectively).**

A   The amino acid change Arg251Gly in the RP2 protein was predicted as damaging according to SIFT and PolyPhen thresholds. The original residue (Arg251) of α-helix forms a hydrogen bond with the Ser219 and Ile220; however, the new residue is highly destabilizing. Specifically, the new residue (Gly) is uncharged, more hydrophobic and smaller than the original, which causes that the positive charge will be lost and the amino acid will not be in the correct position, hampering the establishment of the original hydrogen bond.

B   The amino acid change Arg48Lys in the LXN protein was classified as non-damaging according to the criteria used. The new amino acid, whose substitution was predicted as non-damaging by SIFT and Polyphen, does not cause a significant change in protein stability, maintaining the same charge and polarity as the wild-type residue.

In order to understand the basis of the robustness of the interactome against the deleterious variants carried by normal individuals, the normal interactomes were compared with simulated interactomes in which the same number of damaged proteins was randomly removed (see Materials and Methods). The comparison between the real and simulated interactomes resulted in significant differences between them in the network parameters measured. Real normal populations (1,000 genomes, Spanish population and CLL germinal line) always have more connections than simulated individuals (compare *real populations* bar to *simulated populations with uniform probability* bar in Fig 5A). Moreover, these connections preserved in real individuals are organized in a way that maintains a significantly lower average length of shortest paths (same comparison in Fig 5B), a distinctive feature of biological networks, and avoids disconnection from the giant component (same comparison in Fig 5C). In other words, real individuals have significantly more structured and less affected interactomes than simulated individuals for the same number of removed (damaged) proteins. The results

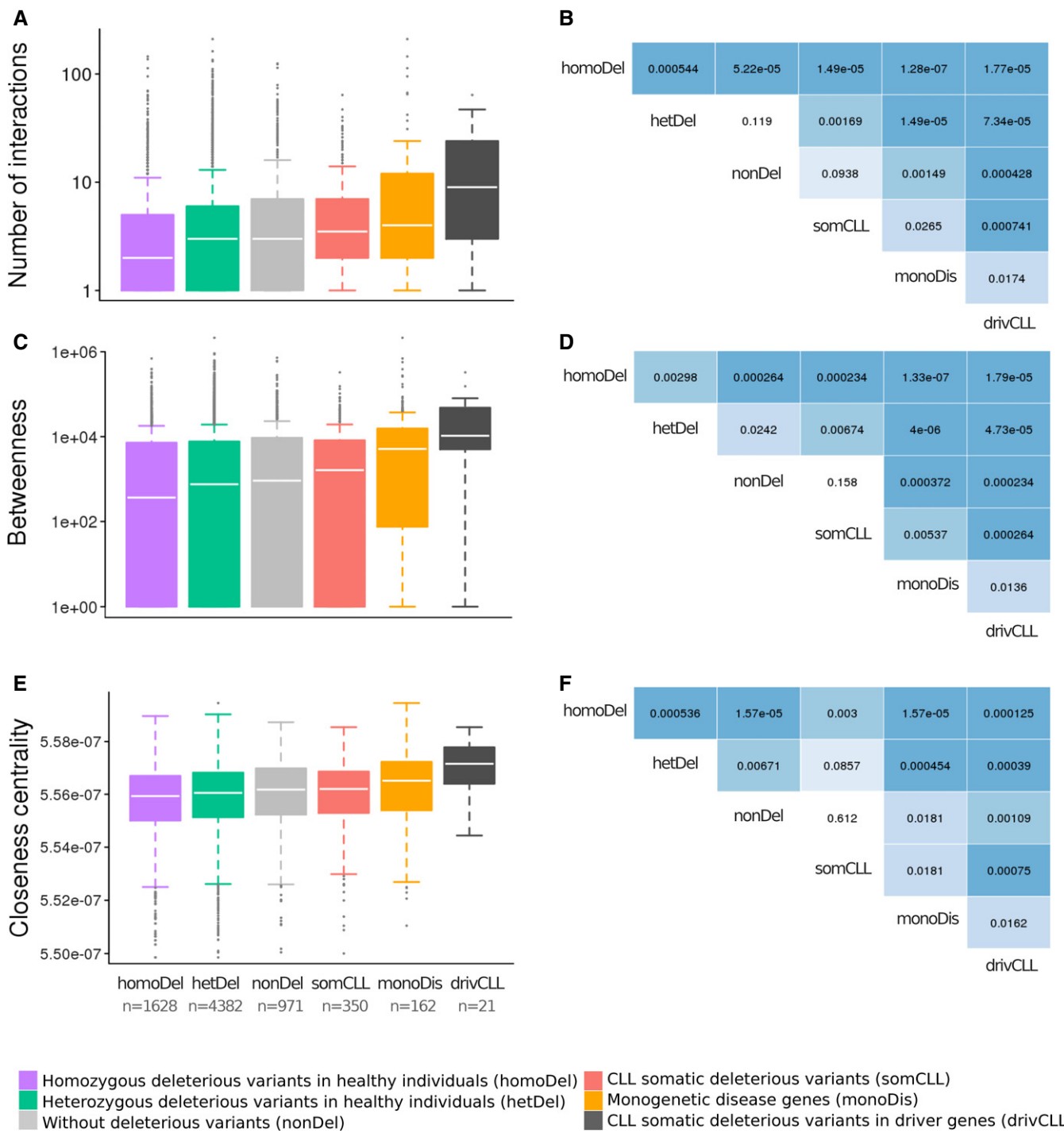

**Figure 3.   Connection degree, betweenness and closeness centrality of proteins affected by deleterious variants.**

A   From left to right: Number of interactions in proteins affected by deleterious variants in both alleles (homozygosis), in only one allele (heterozygosis), not affected by any deleterious variant, proteins affected (in homozygosis or heterozygosis) in a pathological condition (somatic variants in CLL), proteins affected by monogenic diseases (listed in Supplementary Table S2) and the subset of somatic variants in CLL which occur in cancer driver proteins (Vogelstein *et al*, 2013) (listed in Supplementary Table S3).

B   Significance of the comparisons tested by the rank sum (Mann–Whitney *U*-test) with FDR multiple testing adjustments.

C   Betweenness in the same groups of proteins as in (A).

D   Significance of the comparisons tested as in (B).

E   Closeness centrality in the same groups of proteins as in (A).

F   Significance of the comparisons tested as in (B).

Data information: In this boxplot representation boxes correspond to 50% of the observed values and error bars to 90%.

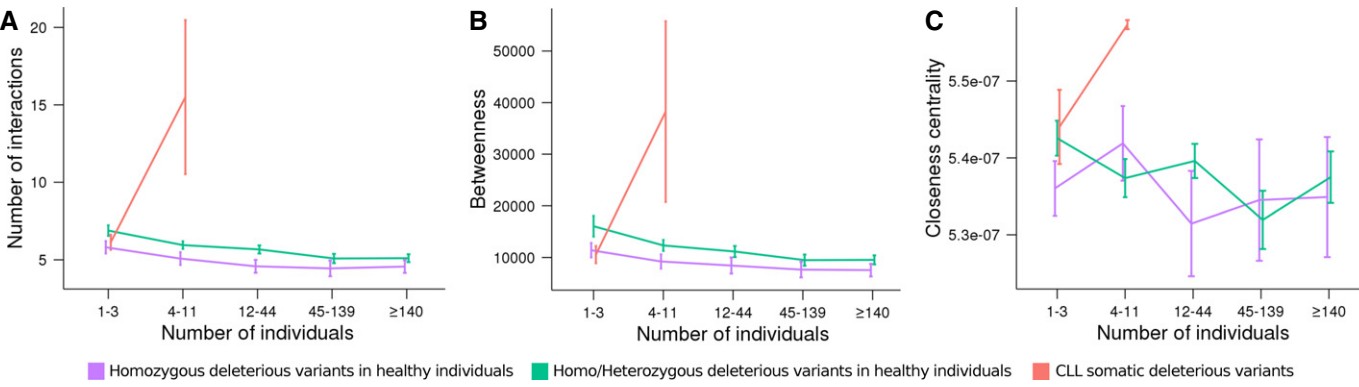

**Figure 4.   Mean connectivity (A), betweenness (B) and closeness centrality (C) for proteins undergoing deleterious variants.**

A–C   The purple line represent deleterious variants in both alleles (homozygosis) and the green line deleterious variants in at least one allele (homozygosis+heterozygosis), grouped according to the number of individuals in normal populations (1,000 genomes and Spanish populations) in which they were observed. The red line represents CLL somatic heterozygous deleterious variants observed in growing number of individuals (within the sample of patients). The plots include 1SD bars. Error bars indicate the dispersion of values observed across the individuals analysed.

were highly significant for the 1,000 genomes population and still significant but with higher *P*-values for the MPG and CLL populations, due to the smaller sample sizes (see *P*-values in Fig 5A–C).

This simulation demonstrates that healthy individuals carry deleterious variants in a specific set of proteins, the deletion of which has minimal impact on the interactome structure. However, it is not clear whether this low impact is due to the actual individual proteins observed in the population or whether it occurs because proteins with deleterious variants are only tolerated in specific combinations which minimize the damage to the interactome structure. To address this question, another simulation was conducted in which deleterious mutations were assigned to proteins according to their observed mutation frequencies in healthy individuals (1,000 genomes and MGP1 populations). In contrast to the previous simulation, the simulated individuals only carried deleterious variants in proteins which are affected in normal individuals, but in random combinations which do not necessarily exist in real healthy individuals.

Although not as remarkable as in the previous simulation, the difference between real and simulated values was also significant. Again, real normal populations had significantly more connections than simulated individuals (compare real populations bar to simulated populations with observed frequencies bar in Fig 5A). These connections result in a network with shorter shortest paths between components (see how average lengths of shortest pathways change across real and simulated populations in Fig 5B) and have a tendency to display fewer isolated components (same comparison in Fig 5C). The *P*-values were higher and in some cases non- significant (number of components for MGP1 and CLL germinal populations, probably due to their small sizes), as the effect of removing the acceptable combination of damaged proteins is not as strong as the effect of removing random proteins.

Collectively, the results obtained suggest that only a limited number of variants in specific combinations are tolerated by the interactome and are compatible with a healthy condition.

Some examples visually illustrate the type of connections lost in the simulation with random occurrences of deleterious variants

when compared to the type of connections lost in the case of observed deleterious variations. Figure 5D depicts an example of sub-networks disconnected from the interactome of a normal individual from the 1,000 genomes project, because both alleles of the gene coding the connecting protein had deleterious variants. Figure 5E shows an example taken from a simulated individual. It is clear that while interactomes of real individuals are slightly trimmed off by the deleterious variants they carry, the interactomes of simulated individuals undergo more serious damage and have larger disconnected portions.

**Deleterious variants observed in normal individuals tend to occur at the periphery of the interactome**

In order to understand the reasons why such specific combinations of deleterious variants cause both minimal disruption to the interactome and are not associated with pathological effects, their location within the network of protein interactions was examined. Firstly, a summarized representation of the interactome was derived by detecting neighbourhoods of densely connected sub-graphs which define communities, or modules of highly interacting proteins (Pons & Latapy, 2005; Rosvall & Bergstrom, 2008). These modules can be considered functional entities which enable the biological interpretation of the results. Then, the distribution of genes carrying alleles affected by deleterious variants across the modules was studied in individuals from the Spanish population and the 1,000 genomes populations.

The pattern of distribution of affected modules across populations is defined by conventional hierarchical clustering using the Euclidean distances between them. The clustering obtained was quite coherent with the geographical origins and history of the analysed populations (Fig 6). The Spanish population is located close to the rest of the European populations as well as to Latin American populations, with whom they share some common background. The deleterious germinal variants found in CLL patients are located close to the Spanish population, probably because it is mainly composed of Spanish CLL patients. On the contrary, the distribution of mutations of somatic deleterious mutations of CLL (Fig 6) follows

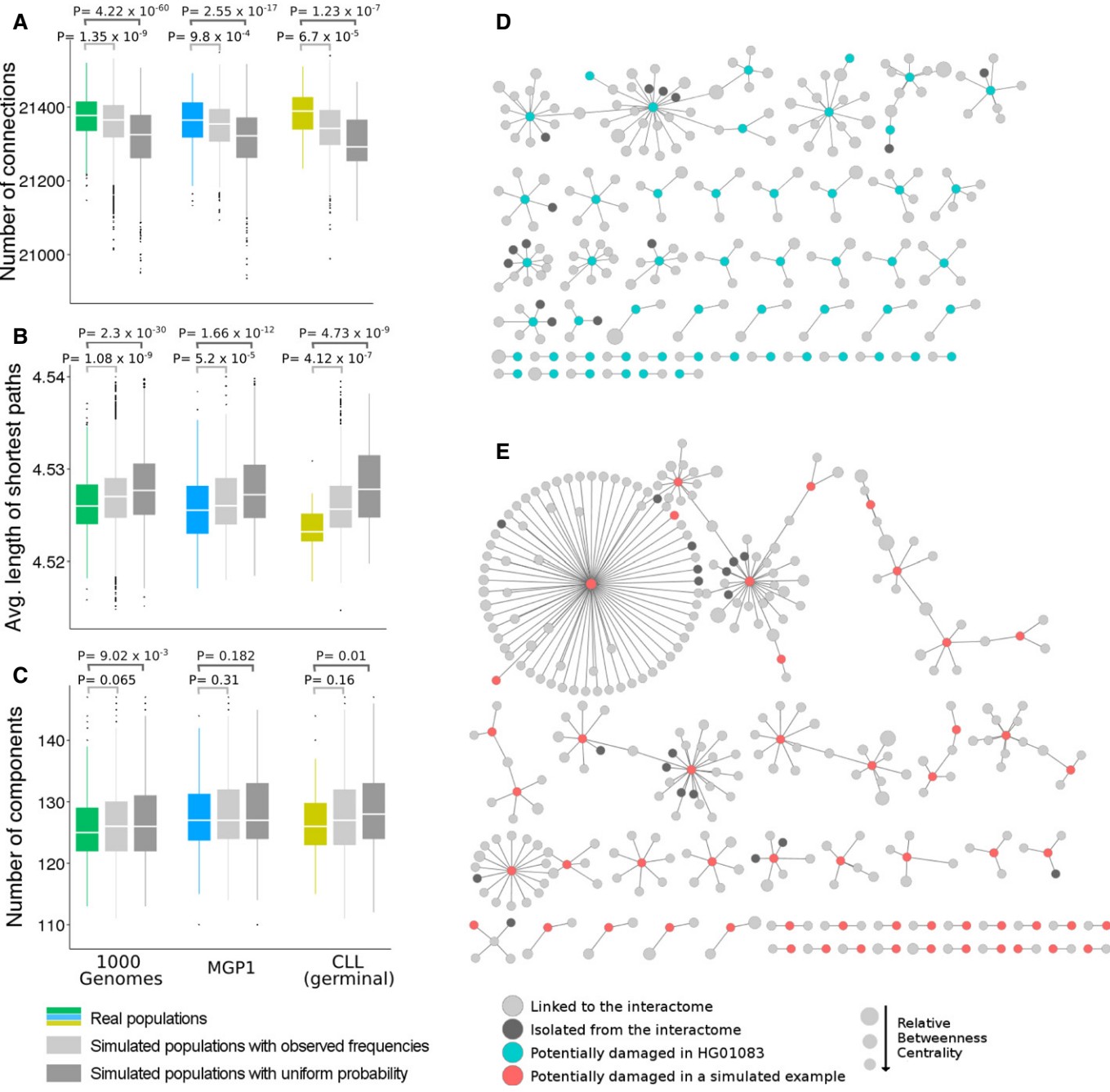

**Figure 5.   Impact of potentially deleterious variants on the interactome of real and simulated individuals.**

A–C   Comparison of the interactome damage between real and random individuals after removing the nodes corresponding to proteins containing deleterious variants in both alleles (homozygote). Two different scenarios are simulated: *Simulated populations with uniform probability*, where proteins are randomly removed, and *Simulated populations with observed frequencies*, where proteins are removed with a probability proportional to the frequency of variation in the 1,000 genomes population (see Results). The comparison was performed using all the 1,000 genomes project populations (green box), the newly sequenced Spanish population MPG1 (blue box) and the germinal variants of the CLL patients (yellow box) and contrasting their distributions with the corresponding random distribution (grey boxes). The effects on the global network topology were defined by (A) the number of connections in the remaining interactome, (B) the average length of the shortest paths and (C) the total number of isolated components. *P*-values are provided for the comparisons of both simulation scenarios with real values.

D,E   Visual illustration of the network components lost after removing nodes corresponding to damaged proteins in (D) a real individual from 1,000 genomes (HG01083 of the PUR population) and (E) a simulated individual with the same number of damaged proteins (i.e. nodes removed).

a pattern inverse to the rest of the normal populations. This anomalous distribution clusters this sample outside of any human population.

The same clustering methodology was applied to group the modules. The analysis resulted in the definition of five main clusters. The two clusters at the bottom are composed of highly affected

modules, enriched in proteins with deleterious variants. The central cluster is composed of protected modules, with a lower proportion of proteins with deleterious variants than expected by chance. The two upper clusters correspond to an intermediate situation.

The distribution of cell functionalities across the modules is depicted in Fig 6. The cluster containing protected (and often central) modules is enriched in GO terms related to essential cellular functions, such as gene expression, translation, protein targeting and chromatin organization. Conversely, the most external clusters contain cell functionalities acquired later in evolution, mainly related to signalling immune response and cell communication (central and part of the upper clusters in Fig 6). Supplementary Table S4 lists a detailed enrichment analysis of all the modules which confirms the observation made.

Then, the distribution of affected proteins across interactome modules in the individuals was analysed. It was observed that modules located at the periphery of the interactome are considerably enriched in affected proteins, while the opposite tendency is observed in internal modules (as portrayed in Fig 7A). The extent of this trend is confirmed by the significant negative correlation (Spearman's correlation test $P$-value $\leq 0.001$) of a measure which accounts for the centrality of a module in the interactome (closeness centrality) with the normalized proportion of affected proteins with respect to the random expectation (relative damage of the module) (Fig 7B). In order to check whether this observation was reflecting the centrality of individual proteins or whether it was accounting for the centrality of the modules, the data were reanalysed using only the centrality of each protein within the context of the network. The interactome was divided into four regions according to the closeness centrality distribution quartiles, and the distribution of damaged proteins among the four regions was calculated for each individual. The result obtained was the same: the peripheral regions of the interactome accumulated more proteins affected by deleterious mutations than expected by chance, whereas the internal region displayed a remarkable reduction ($P$-value $= 3.96 \times 10^{-6}$ Mann–Whitney $U$-test) in affected proteins (see Supplementary Fig S2A and B).

---

**Figure 6.  Heatmap depicting the distribution of the proteins harbouring deleterious variants in at least one allele across the interactome modules in the 1,000 genomes populations, the Spanish MGP1 population, germinal and somatic CLL.**

Rows represent the different interactome modules defined by the Waltrap clustering algorithm, which are labelled with the corresponding identification number. The columns represent the populations analysed. The colour code represents the relative damage of the module, which accounts for the deviation in the proportion of affected proteins in the module from the random expectation distribution. Red indicates samples presenting more affected proteins than the random expectation, whereas blue indicates a negative difference. There are five main clusters defined by conventional hierarchical clustering using the Euclidean distances between the rows. On the right of the figure, the main GO terms which are significantly enriched in any of the mains clusters are displayed. The left column corresponds to cellular component ontology and the right column to the biological process ontology. When columns are clustered, the groups obtained according to the similarities in the distribution of affected proteins across modules correspond exactly to the geographical localization of the populations. The distribution of somatic deleterious mutations of CLL follows a pattern inverse to the rest of the normal populations.

Source data are available online for this figure.

## Germinal and somatic, cancer-specific mutations in CLL

Here, the focus is on comparing the distribution of deleterious variants in genes across the different communities in both the germinal line (which would represent a normal genome) and somatic mutations in the cancer samples (corresponding to a pathological condition) of CLL patients. The germinal line of CLL patients presents a pattern of distribution of variants indistinguishable from normal individuals, clustering close to the Spanish population (Fig 6). However, the pattern of somatic mutations in CLL is completely different to any other population and is actually inverted to the pattern observed in normal individuals. Figure 7C documents the inverse trend of distribution of mutations when represented on the interactome of modules. As opposed to the case of normal populations (Fig 7B), deleterious somatic

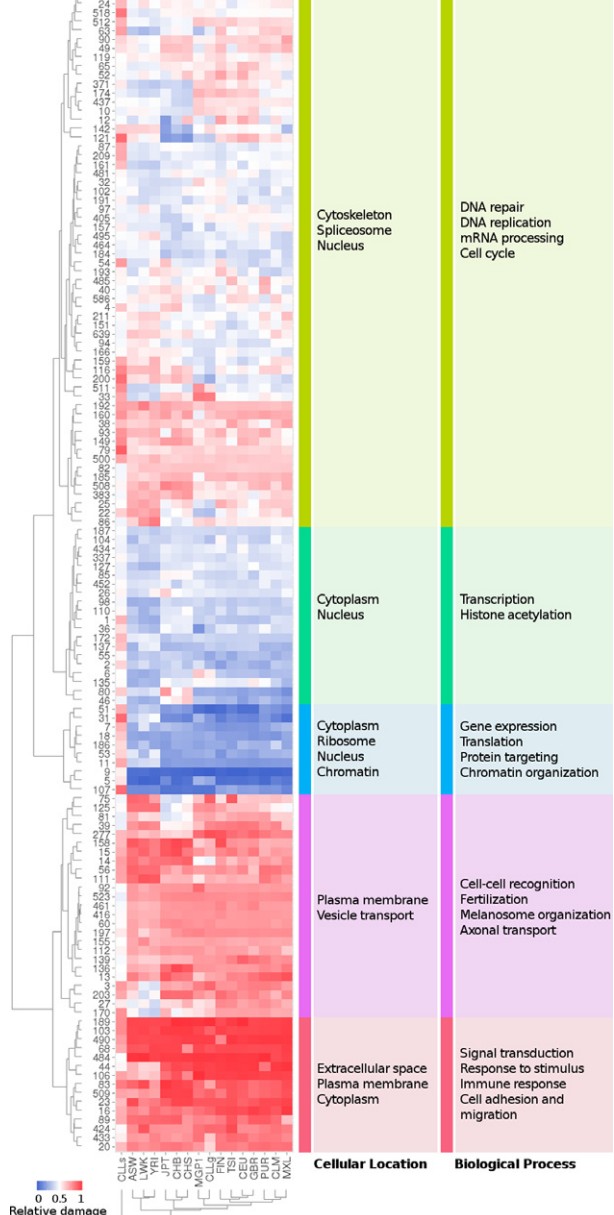

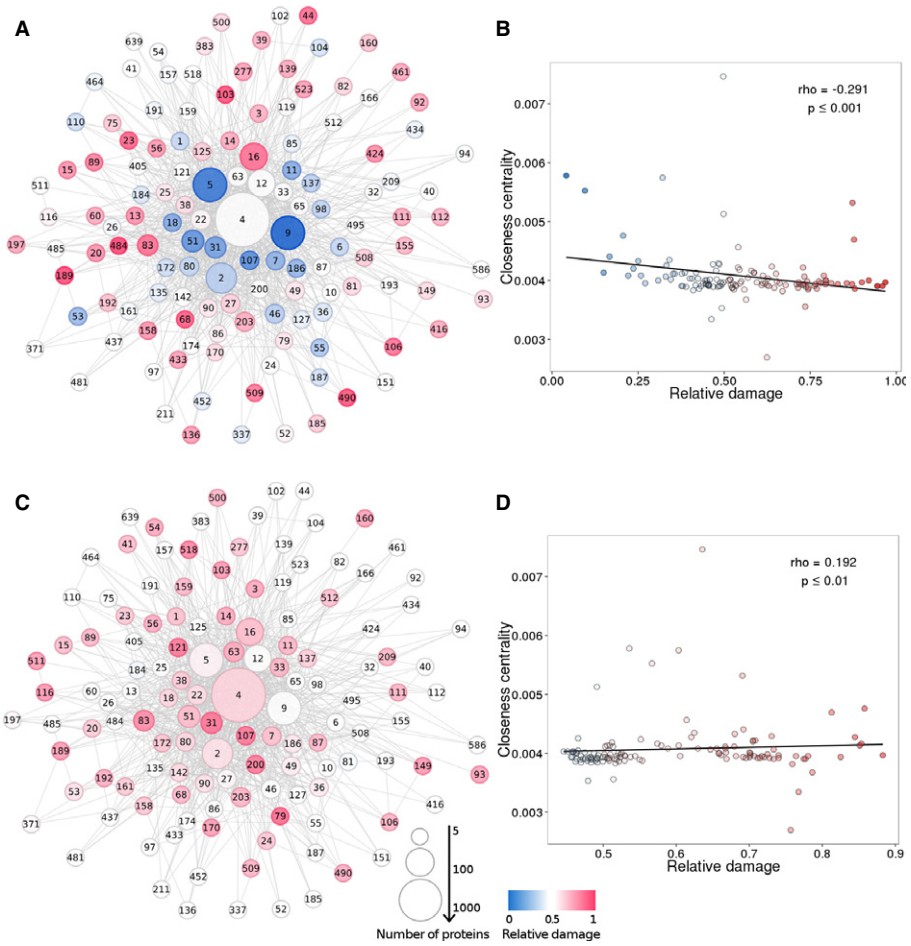

**Figure 7.   Interactome distribution of the deleterious variants in normal populations, and also within germinal and somatic variants in CLL patients.**

Network of interactome modules as defined by the *Walktrap* clustering algorithm. Two modules are connected if there is at least one interaction between one of their respective proteins. The numbers in the nodes are the identifiers of each module. The size of the node is proportional to the number of proteins which it contains. The colour code represents the relative damage value. Relative damage values range between 0 (no proteins affected at all in this module) and 1 (the maximum possible number of proteins affected in this module). Blue indicates that the frequency of damaging variants is below the median in the simulated individuals (which would correspond to a value of 0.5), whereas red indicates that the value is above the median.

A   Distribution of proteins with deleterious variants in the 1,078 individuals from the 1,000 genomes populations plus the 252 individuals in the MPG1 Spanish population and the 41 germinal CLL exomes across the interactome of modules.

B   Closeness centrality, which gives a measurement on how central a node is (the higher, the more central) as a function of the relative damage of the module. There is a significant trend of accumulation of mutations as modules are more peripheral.

C   Distribution of proteins with deleterious variants in the 41 somatic CLL exomes, representative of a pathological condition, across the interactome of modules.

D   Closeness centrality as a function of the relative damage of the module for the 41 somatic CLL exomes. Opposite to the above, there is a significant trend of accumulation of mutations as modules are more central.

Source data are available online for this figure.

mutations in CLL are over-represented in internal modules of the interactome. The significance of this trend is confirmed by the significant positive correlation (Spearman's correlation, $P$-value ≤ 0.01) existent between a measure of the module centrality within the interactome (closeness centrality) and the proportion of affected proteins with respect to the random expectation (relative damage of the module) (Fig 7D). The opposite trends observed both in normal populations and in somatic mutations of CLL (see Table 2) patients have been confirmed using different interactomes and different algorithms for defining modules within them (see Table 3).

# Discussion

Different genomic initiatives (The_Cancer_Genome_Atlas_Research_Network, 2008; Durbin *et al*, 2010; Hudson *et al*, 2010; Dunham *et al*, 2012) as well as an increasing collection of genomes of patients affected by rare diseases (Ng *et al*, 2010) are producing a fast-growing catalogue of variants in the human genome. Among these, an unexpectedly high number of LoF variants, with no apparent phenotypic consequence, have been discovered in healthy human populations (MacArthur & Tyler-Smith, 2010; Nothnagel *et al*, 2011; Xue *et al*, 2012). Consequently, there is an increasing

**Table 2.** Spearman's rank correlation coefficient (ρ) between the number of interactions, the betweenness and the closeness centrality with respect to the frequency of occurrence of deleterious variants in the population in three different scenarios: homozygosis, heterozygosis and the somatic mutations observed in the CLL patients.

| | Number of interactions | | Betweenness | | Closeness centrality | |
|---|---|---|---|---|---|---|
| | ρ | *P*-value | ρ | *P*-value | ρ | *P*-value |
| Homozygosis | −0.0391 | 0.101 | −0.0341 | 0.154 | −0.0238 | 0.319 |
| Heterozygosis | −0.0661 | $1.34 \times 10^{-7}$ | −0.0534 | $2.053 \times 10^{-5}$ | −0.0536 | $1.934 \times 10^{-5}$ |
| CLL somatic | 0.152 | 0.0116 | 0.118 | 0.051 | 0.128 | 0.0335 |

**Table 3.** Validation of the relationship between the module centrality and damage using different network module detection algorithms (*Infomap* and *Walktrap*) and three protein interactomes (see Materials and Methods).

| Sample | Interactome | Network module detection algorithm | Rho | *P*-value |
|---|---|---|---|---|
| 1,000 genomes, MGP1 and germinal CLL | Curated | *Walktrap* | −0.292 | ≤ 0.001 |
| | Curated | *Infomap* | −0.159 | ≤ 0.001 |
| | Non-Curated | *Walktrap* | −0.13 | 0.28 |
| | Non-Curated | *Infomap* | −0.11 | ≤ 0.01 |
| | STRING | *Walktrap* | −0.186 | ≤ 0.01 |
| | STRING | *Infomap* | −0.205 | ≤ 0.01 |
| Somatic variants CLL | Curated | *Walktrap* | 0.192 | ≤ 0.01 |
| | Curated | *Infomap* | 0.176 | ≤ 0.001 |
| | Non-Curated | *Walktrap* | 0.321 | ≤ 0.01 |
| | Non-Curated | *Infomap* | 0.211 | ≤ 0.01 |
| | STRING | *Walktrap* | 0.28 | ≤ 0.00 |
| | STRING | *Infomap* | 0.322 | ≤ 0.001 |

need to distinguish variants that correspond to polymorphisms in human populations, and especially LoF variants, from those causative of diseases.

The concept of the pathological effect of a variant has traditionally been considered an intrinsic property of the protein. Many methods have been proposed to predict the potential pathological consequences of variants (Ng & Henikoff, 2001; Stone & Sidow, 2005; Arbiza *et al*, 2006; Conde *et al*, 2006; Reumers *et al*, 2008; Kumar *et al*, 2009; Adzhubei *et al*, 2010; Goode *et al*, 2010; Gonzalez-Perez & Lopez-Bigas, 2011). However, with some exceptions, all these tools predict a pathological effect with an accuracy of between 70% and 80% (Gonzalez-Perez & Lopez-Bigas, 2011). This suggests that observing the occurrence of a damaging variant in a protein is a necessary, although not sufficient, condition for it to have a pathological effect.

Here, instead of studying the pathological effect of deleterious variants in the context of disease, as in many previous works (Ng & Henikoff, 2001; Stone & Sidow, 2005; Arbiza *et al*, 2006; Conde *et al*, 2006; Reumers *et al*, 2008; Kumar *et al*, 2009; Adzhubei *et al*, 2010; Goode *et al*, 2010; Gonzalez-Perez & Lopez-Bigas, 2011), a radically different approach was followed. Taking advantage of the availability of a wealth of genomes of healthy individuals, the

reasons why apparently deleterious variants cause no pathological effect in them were analysed. Given the modular nature of human genetic diseases (Brunner & van Driel, 2004; Gandhi *et al*, 2006; Lim *et al*, 2006; Goh *et al*, 2007; Oti & Brunner, 2007; Wagner *et al*, 2007; Oti *et al*, 2008), and the trend of disease genes to reside in a neighbourhood within the network of protein interactions (Lim *et al*, 2006; Lage *et al*, 2007; Ideker & Sharan, 2008; Vidal *et al*, 2011; Mitra *et al*, 2013), the interactome was used in the present study as a scaffold connecting proteins in a way related to common functionality. It is known that experimental artefacts, limitations in screening power and sensitivities of particular assays can yield false positives in interaction data (von Mering *et al*, 2002). To avoid potential artefacts and false positives, only interactions detected by at least two different detection methods were used. In this way, a high-quality curated interactome was produced with which the effect of deleterious mutations could be studied. In this study, deleteriousness was defined on the basis of the pathogenicity scores SIFT (Kumar *et al*, 2009) and Polyphen (Ramensky *et al*, 2002). Since both scores are known to sometimes produce discrepant results (Hicks *et al*, 2011), their validity was checked with an *in silico* study which resulted in a reasonable agreement between the prediction of the deleterious effect of an amino acid change and its impact on the protein structure.

This work provides an explanation for the maintenance of a seemingly high mutational load in healthy individuals. The individualized observations made in healthy subjects and CLL patients, completed with the analysis of proteins mutated in monogenic diseases, strongly suggest that the pathogenic role of deleterious mutations is highly correlated with the impact on the interactome integrity caused by the combined LoF of the affected proteins, which is also related to the location of such proteins within the interactome. Thus, affected proteins in healthy individuals are concentrated in peripheral modules, avoiding internal modules. However, the most important factor which sheds light on the mechanisms by which the interactome can bear a large number of proteins with deleterious mutations is related to the way in which affected proteins are specifically combined in healthy individuals. Affected proteins in healthy individuals tend to occur in combinations which preserve short path lengths (Fig 5B). When the same proteins occur in random combinations, the length of the shortest paths significantly increases (Fig 5B). Most probably, the structural constraints imposed by the preservation of shortest paths underlie the relative higher tolerance for deleterious mutations observed in the periphery of the interactome. In the periphery, combinations of affected proteins that preserve shortest path lengths are easier to find than in internal regions of the interactome. Visually, the effect on the interactome caused by removing such combinations of damaged proteins seems

to be restricted to the disconnection of some very small marginal components, often composed of a few proteins, as opposed to the effect observed when the affected proteins are removed randomly (compare Fig 5 D and E). This property could only be observed by means of an individualized analysis of the healthy subjects.

In addition, the results presented here are in agreement with previous studies which report that proteins involved in genetic diseases show little preference for either the centre or the periphery of the interactome (Goh *et al*, 2007) and situate cancer proteins in a central location of the interactome (Jonsson & Bates, 2006; Rambaldi *et al*, 2008; Vidal *et al*, 2011) (see Fig 7 and Supplementary Fig S2). Thus, as previously suggested, loss of phenotypic robustness might be a phenomenon that occurs when cellular networks are disrupted (Levy & Siegal, 2008).

Despite being carefully curated, the use of the interactome always entails a risk of obtaining results biased towards well-studied biological processes (Edwards *et al*, 2011; Das & Yu, 2012). Thus, the true degree of understudied proteins could be underestimated in comparison with that of the well-studied proteins. It might be argued that this effect could, for example, inflate the differences in network parameters between cancer genes and other classes. Nevertheless, the interactome used here is expected to suffer to a lesser extent from this bias, given that protein interaction data were only retrieved from raw data repositories, avoiding knowledge-based sources.

From an evolutionary perspective, the interactome seems to have grown by the addition of external components rather than by radical internal re-structuring. Actually, our results (Supplementary Fig S1) agree with previous observations which document how proteins under positive selection tend to be placed at the periphery of the interactome, whereas proteins under negative selection tend to have a central location in the interactome (Fraser *et al*, 2002; Kim *et al*, 2007). This observation has been extrapolated to functional modules (Serra *et al*, 2011), which overlap with network modules (Minguez & Dopazo, 2011) or communities to some extent. From the point of view of disease, a high degree of connectivity between proteins mutated in the same disease state has been reported (Goh *et al*, 2007). This observation suggests that interaction modules carry out functions which can be impaired by the failure of one or several of their nodes.

The general conclusion of this work is that the deleterious character of a variant obviously depends on the damage it causes to the protein, but ultimately, it is a system's property that critically depends on the location of the affected protein within the interactome and, especially, on the relative location of the specific combination of affected proteins within the interactome. Deleterious variants affecting genes of internal interactome modules will probably disrupt the network structure and affect more essential functionalities. Consequently, they will likely have pathological consequences. On the other hand, deleterious variants which affect specific combinations of proteins in peripheral modules of the network in a way that minimizes the increase of shortest paths and, consequently, the loss of interconnectivity have a high likelihood of both causing minor distortions to the interactome and affecting only non-essential functionalities. Variants of this type are observed in normal individuals and have little or no pathological consequences. Moreover, this work stresses the importance of the analysis not only of the diseased condition but also of the healthy condition when examining the consequences of genomic features.

# Materials and Methods

### Interactome data

Protein–protein interactions were obtained from the following databases: BioGRID (Chatr-Aryamontri *et al*, 2013) version 3.1.89 downloaded on 17 April 2012; IntAct (Kerrien *et al*, 2012) released on 17 April 2012; and Molecular Interaction Database (MINT) (Licata *et al*, 2012) released on 2 December 2011. Data were processed as follows: (i) only proteins with UniProt Swiss-Prot (UniProt_Consortium, 2011) IDs were used; (ii) only interactions of 'physical association' type were used; and (iii) only interactions detected by at least two different detection methods (von Mering *et al*, 2002) were used. The subset of interactions obtained after these filtering steps constitute a curated interactome which comprises a total of 7,331 proteins connected by 21,623 interactions. The categories 'physical association' and 'detection method' are components of the xml format PSI-MI 2.5 (Kerrien *et al*, 2007) offered by the PPI databases used. Additionally, two other protein interactomes were considered. One of them was built by including every binary interaction from the above databases, which fits the previous criteria with the exception of iii (non-curated interactome), containing 82,852 binary curated interactions between 12,118 proteins. The other was built by including 52,726 binding interactions between 10,662 proteins with a score higher than 400 from the STRING database.

### Human populations

A total of thirteen human populations were used in this study which include European populations TSI from Tuscany in Italia (98 samples), FIN Finnish from Finland (93 samples), GBR British from England and Scotland (89 samples), CEU which are Utah residents (CEPH collection) with northern and western European ancestry (85 samples); Asian populations CHB Han Chinese in Beijing, China (97 samples), CHS Han Chinese South (100 samples) and JPT Japanese in Tokyo, Japan (89 samples); American populations MXL Mexican Ancestry in Los Angeles, CA (66 samples), PUR Puerto Rican in Puerto Rico (55 samples) and CLM Colombian in Medellin, Colombia (60 samples); and African populations YRI Yoruba in Ibadan, Nigeria (88 samples), LWK Luhya in Webuye, Kenya (97 samples), and ASW African Ancestry in southwest USA (61 samples). The exome sequences of all the individuals corresponding to the thirteen populations were downloaded from the 1,000 genomes web page (http://www.1000genomes.org/) in multi-sample VCF format. Variants in positions located in the interactome genes were collected for this study (see below).

This selection was completed with MGP1, a population composed of 252 Spanish samples from healthy individuals, sequenced in the context of the Medical Genome Project (http://www.medicalgenomeproject.com). The total number of individuals studied in all the populations is 1,330. Finally, 41 exomes of chronic lymphocytic leukaemia (CLL) patients (Quesada *et al*, 2012) were analysed.

## Human subjects

The exome sequences of the human populations described above (except MGP1) were downloaded from the 1,000 genomes web page (http://www.1000genomes.org/) in multi-sample VCF format (February 2012 release).

Following informed consent, the 252 MGP1 samples were obtained and further anonymized and sequenced.

Collection of samples from patients and their use for research were ethically approved by the University Hospital Virgen del Rocío (Seville, Spain) institutional review board for the protection of human subjects and performed according to the principles set out in the WMA Declaration of Helsinki.

Sequence data have been deposited at the European Genome-phenome Archive (EGA), which is hosted by the EBI, under accession number EGAS00001000938. The exomes of 41 CLL patients (Quesada *et al*, 2012) were downloaded from the EGA repository (ID: EGAD00001000044).

## Construction of DNA libraries and sequencing

Library preparation and exome capture were performed according to a protocol based on the Baylor College of Medicine protocol version 2.1 with several modifications. Briefly, 5 µg of input genomic DNA is sheared, end-repaired and ligated with specific adaptors. A fragment size distribution ranging from 160 bp to 180 bp after shearing and 200–250 bp after adaptor ligation was verified by Bioanalyzer (Agilent). The library is amplified by pre-capture LM-PCR (linker-mediated PCR) using FastStart High Fidelity PCR System (Roche) and barcoded primers. After purification, 2 µg of LM-PCR product is hybridized to NimbleGen SeqCap EZ Exome libraries V3. After washing, amplification was performed by post-capture LM-PCR using FastStart High Fidelity PCR System (Roche). Capture enrichment is measured by qPCR according to the NimbleGen protocol. The successfully captured DNA is measured by Quant-iT™ PicoGreen dsDNA reagent (Invitrogen) and subjected to standard sample preparation procedures for sequencing with SOLiD 5500xl platform as recommended by the manufacturer. Emulsion PCR is performed on E80 scale (about 1 billion template beads) using a concentration of 0.616 pM which contains four equi-molecular pooled libraries of enriched DNA. After breaking and enrichment, about 276 million enriched template beads are sequenced per lane on a 6-lane SOLiD 5500xl slide.

## Analysis of the Spanish population and the CLL sequencing Data

A customized pipeline was applied for processing the sequences. In brief, sequence reads were aligned to the reference human genome build GRCh37 (hg19) using the SHRiMP tool (Rumble *et al*, 2009). Reads correctly mapped were further filtered with SAMtools (Li *et al*, 2009), which was also used for sorting and indexing mapping files. Only high-quality sequence reads mapping to the reference human genome in unique locations were used for variant calling. The Genome Analysis Toolkit (GATK) (McKenna *et al*, 2010) was used to realign the reads around known indels and for base quality score recalibration. Identification of single nucleotide variants and indels was performed using GATK standard hard filtering

parameters (DePristo *et al*, 2011). In the case of CLL samples, the calling of somatic variants was carried out with the specialized software Mutect (Cibulskis *et al*, 2013).

## Selection of deleterious variants

Firstly, the functional consequence of every variant was computed using VARIANT (Medina *et al*, 2012) software and those affecting either the protein sequence or the mRNA transcription/translation were selected. Variants located in intronic, upstream, downstream or intergenic regions, as well as variants with synonymous or unknown functional consequence, were filtered out. Only non-synonymous, stop loss, stop gain and splicing disrupting variants were considered. Then, the putative impact and damaging effect of these variants on the functionality of the affected protein was predicted by computing both SIFT (Kumar *et al*, 2009) and Poly-phen (Ramensky *et al*, 2002) damage scores. This was completed with phastCons (Siepel *et al*, 2005) conservation score. Since the conservation score is the only parameter applicable to any type of position, it was used as a primary filter. Thus, stop loss, stop gain and splicing disrupting variants with a phastCons conservation score higher than 200 were selected as damaging. In the case of non-synonymous variants, a SIFT score lower than 0.05 or a Poly-phen score higher than 0.95 is also required to consider them as deleterious.

## Source of disease annotations

A total of 1,746 uniprot-OMIM disease terms associations were downloaded from the UniProtKB/Swiss-Prot database (release April 2014). The Disease Ontology (Schriml *et al*, 2012) was used to classify OMIM disease terms. Those proteins associated with OMIM terms annotated under the disease ontology parent 'monogenic disease' (DOID:0050177) were tagged as monogenic disease-associated proteins, comprising a total of 162 uniprot accessions in our curated interactome (see Supplementary Table S2). Cancer driver genes (a total of 138) were taken from the study by Vogelstein *et al* (2013) (see Supplementary Table S3).

## *In silico* structural analysis of the impact of mutations in the proteins

Protein sequences were downloaded from the UniProt database (The_Uniprot_Consortium, 2014). Only proteins structurally solved in the PDB (Berman *et al*, 2000) were used here for validation. Three-dimensional models were produced for each protein using the RaptorX program (Kallberg *et al*, 2012). The program performs a template-based protein structure modelling, applying single- and multiple-template threading methods. The three-dimensional model was used to predict the effect that single point mutations have on the stability of proteins, using SDM software (Worth *et al*, 2011). SDM calculates a stability score which accounts for the free energy difference between the wild-type protein and the corresponding mutated protein. Additionally, some sequence-based features, such as changes in the charge and the polarity of the protein, as well as SNAP predictions (Bromberg & Rost, 2007), were used to further assess the severity of the impact produced by the change. Changes in charge and polarity were defined exclusively on the

basis of the type of residue substitution. Changes in polarity and charge were based uniquely on the residue changed. Polarity changes were measured on a hydrophobicity scale of 0 (LIFWCMVY), 1 (PATGS) or 2 (HQRKNED) (Mirkovic *et al*, 2004). Changes in the total protein charge were estimated on the basis of the charges of the residues: positive (RK), negative (ED) or non-charged (LIFWCMVYPATGSHQN).

## Assessing the effect caused by homozygote deleterious variants on the interactome structure

The aim of this study was to quantify the global damage that the deleterious variants cause to the interactome. To achieve this, individual interactomes were constructed by removing those nodes affected by homozygote deleterious variants from the network of protein interactions. Then, the impact that this subtraction of nodes had on the interactome structure was studied. In addition to homozygote deleterious variants, the cases of proteins with compound heterozygote alleles were also removed. In particular, the impact on the interactome was assessed by measuring the following network properties: (i) separation into isolated components, via the total number of components or the size of the giant component; (ii) connectivity loss: via the total number of remaining edges; and (iii) increase of path lengths, by measuring the network diameter (largest shortest path) or the average path length.

Then, the aim was to understand the extent of the damage produced by the deleterious variants on the interactomes of real individuals. To evaluate this, the network properties of real individual interactomes were compared with simulated interactomes in which a similar number of affected nodes were randomly removed. In these simulated interactomes, the probability of a protein being affected is identical for any protein in the network. Such simulated interactomes represent the expectation of random damage in the interactome for a given number of affected proteins. These comparisons were performed at population level. Thus, for each population, 1,000 interactomes with a number of affected proteins randomly sampled among the values observed in the population were generated. The average values of network properties of real and simulated interactomes were compared by means of a non-parametric Mann–Whitney test. Another simulation was conducted in which proteins were removed not randomly as before, but rather with a probability proportional to the observed mutation frequencies in the 1,000 genomes and MGP populations. In this scenario, the resulting simulated individuals will have deleterious variants only in proteins which are affected in normal individuals, but in random combinations that do not necessarily exist in real healthy individuals.

Comparison of the observed values of interactome network properties in real individuals with respect to the corresponding distribution of values obtained from the simulated population of interactomes will confirm whether the variants carried by normal population occur in the less damaging positions among all the possible locations or not.

## Defining the modular structure of the interactome

The interactome was divided into communities or modules using the *Walktrap* algorithm (Pons & Latapy, 2005). This algorithm finds densely connected neighbourhoods, also called network communities or modules, within a graph via random walks under the assumption that short random walks are 'trapped' within highly interconnected network regions. A second community detection algorithm, called *Infomap* (Rosvall & Bergstrom, 2008), was used to validate the results. Both algorithms were carried out using the freely available igraph R package (http://cran.r-project.org/web/packages/igraph/), keeping the authors default parameters. In this study, only those communities composed of at least five proteins were used.

## Quantifying the impact in the modules

Once the interactome communities were defined, the distribution of the proteins containing deleterious variants was studied. Here, for every individual, the proportion of affected proteins per module was calculated. To determine how the observed distributions deviate from the random expectations, a permutation test was carried out in which the affected proteins were distributed randomly across the interactome. Again, the probability of a protein being affected in the permutations is the same for any protein in the interactome. Then, empirical random distributions of affected proteins were obtained for each module separately by running 1,000 simulations for each individual. The value of relative damage was defined for each module as the percentile of the empirical random distribution corresponding to the observed proportion of affected proteins in the module. Relative damage values were rescaled between 0 (no proteins affected at all in this module) to 1 (the maximum possible number of proteins affected in this module).

## Description of the communities and human populations based on their damage profiles

Hierarchical clustering on the Euclidian distances based on the comparisons of the module impact values was used to arrange populations according to the resemblance in patterns of impact across communities. In order to gain insight into the biological processes affected or protected across communities, a GO enrichment test of the clusters found was carried out using the FatiGO (Al-Shahrour *et al*, 2004) algorithm as implemented in the Babelomics package (Al-Shahrour *et al*, 2008; Medina *et al*, 2010).

**Supplementary information** for this article is available online: http://msb.embopress.org

## Acknowledgements

We are indebted to Alejandro Aleman for helping with the web interface to the variants. This work is supported by grants BIO2011-27069 and PRI-PIBIN-2011-1289 from the Spanish Ministry of Economy and Competitiveness (MINECO), PROMETEO/2010/001 from the Conselleria de Educació of the Valencia Community. LG-A is supported by fellowship PFIS FI10/00020 from the MINECO. We also thank the support of both initiatives of the ISCIII (MINECO): the National Institute of Bioinformatics (www.inab.org), the CIBER de Enfermedades Raras (CIBERER). We thank the support of Bull through the Bull Chair in Computational Genomics (http://bioinfo.cipf.es/chair_compgenom). We are indebted to the ICGC consortium for making the CLL data used in this study available.

## Author contributions

LG-A did most of the analysis and participated in the writing of the manuscript; LG-A and JJ-A processed the 1,000 genomes samples; LG-A and JC processed the CLL samples; AV, JS and GA produced and processed the Spanish sample data; and JD conceived and coordinated the work and wrote the manuscript.

## Conflict of interest

The authors declare that they have no conflict of interest.

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
