## [Review Process File · Molecular Systems Biology]

The role of the interactome in the maintenance of deleterious variability in human populations

Luz Garcia-Alonso, Jorge Jimenez-Almazan, Jose Carbonell-Caballero, Alicia Vela-Boza, Javier Santoyo-Lopez, Guillermo Antinolo and Joaquin Dopazo

Corresponding author: Joaquin Dopazo, Centro de Investigacion Principe Felipe

Review timeline:	Submission date:	22 February 2014
	Editorial Decision:	27 March 2014
	Revision received:	16 June 2014
	Editorial Decision:	01 August 2014
	Revision received:	23 August 2014
	Accepted:	28 August 2014

Editor: Maria Polychronidou

Transaction Report:

1st Editorial Decision

27 March 2014

Thank you again for submitting your work to Molecular Systems Biology. We have now heard back from the two referees who agreed to evaluate your manuscript. As you will see from the reports below, the referees acknowledge that the presented analysis of the effects of genetic variants on the interactome is potentially interesting. However, they raise a series of concerns, which should be carefully addressed in a revision of the manuscript.

Overall, the reviewers refer to the need to perform additional analyses to more convincingly support the main conclusions. Without repeating all the comments listed below, among the more fundamental points are the following:

- Further analyses are required to examine the generality of the presented conclusions beyond CLL and cancer, considering that, as referee #1 points out, cancer proteins and non-cancer disease proteins differ in their network topology.
- Referee #2 refers to the need to perform additional simulations to better support the conclusions regarding the effect of the variants on the interactome structure.
- Potential biases in the topology of the protein-protein interaction network need to be addressed. Moreover, the generality of the results beyond the presented interactome should be examined and discussed.

If you feel you can satisfactorily deal with these points and those listed by the referees, you may wish to submit a revised version of your manuscript. Please attach a covering letter giving details of the way in which you have handled each of the points raised by the referees. A revised manuscript will be once again subject to review and you probably understand that we can give you no guarantee at this stage that the eventual outcome will be favourable.

REFeree REPORTS

Reviewer #1:

The manuscript presents a carefully elucidated analysis of the effects of genetic variants on protein-protein interactome, through comparing sequence variant data from different populations. This computational analysis attempts to interpret why apparently deleterious variants are carried along in normal, healthy individuals. The sequencing data is used to compare natural variants to variants within chronic lymphocytic leukemia (CLL) patients (somatic and germline mutations). The interactome network they use is derived from literature data taken from curated interaction databases. The major conclusion is that the deleteriousness of a variant depends on the location of the protein in the interactome network. While the question addressed is of wide-reaching interest, many statements are not precisely made, and additional controls or analyses would be required to make the conclusion sound. The text would also need major editing and correction to convey the message optimally.

Major points:

- In the Introduction it is stated that disease proteins tend to be more central in interactome networks. This is actually in contradiction to the papers cited. Since the first demonstration in 2007 (Goh et al., PNAS) and many replications since, it is accepted that disease genes, except for cancer, locate on the periphery of the interactome. Cancer genes are the exception. This is a fundamental problem for this manuscript since the main generalization proposed made is that for all disease genes, cancer or non-cancer, the location of the affected protein in the interactome underlies the deleteriousness of the variant. "Disease proteins" mutated in cancer are more "central" compared to loss-of-function natural variants which are in proteins more on the periphery, but whether this conclusion holds for all disease genes needs to be clarified.

- As a test of the hypothesis that disease mutations are more central in the interactome network somatic mutations from CLL patients are employed. The problem here is that cancer proteins have been shown to differ greatly from non-cancer disease proteins with respect to network topology. To draw more general conclusions proteins associated with diseases other than cancer should also be analysed.

- The PPI networks employed in this work rely on literature-curated interactome (LCI) datasets. Judiciously a high-quality version of the interactome is used by keeping only PPIs supported by several methods. One criticism that has been made about LCI networks is that their topology is systematically biased because some proteins are more studied than others (Das & Yu, BMC Syst Biol 2012; Edwards et al, Nature 2011) It should be checked whether what is observed might be reflecting such bias. At the least discuss this point should be discussed.

- Most of the work relies on the prediction of the deleterious effect of the mutations. The authors use stringent criteria to define such mutations. Assessing experimentally (or for instance using available 3D structures) a subset of these mutations would likely add to this paper.

o Please read the entire manuscript for syntax, grammar and tense mistakes. Some examples are provided here as minor points.

The legends and figures should more precisely indicate what is represented. For instance, is it the "number of mutations" or "average number of mutations per patient"?

Abstract, Last line, "system" should read "systems".

Introduction, Line 4, "Of importance particular" should be "Of particular importance".

Introduction, Paragraph 2, Line 25, "severe disruption" should be only under negative (but not positive) genetic interaction.

Results, second section, the title "Deleterious variants observed in normal population and cancer patients affect to proteins" should be "... affect proteins".

Same section, Paragraph 2, Line 3, "it is still" should be "is still".

Whenever possible, please provide the exact P values, but not just "p-value<0.01".

Reviewer #2:

This manuscript presents a network-based analysis of loss-of-function variants in individual genomes from several populations. The authors provide evidence that the frequency of LoF variants correlate with network properties including connectivity and centrality, suggesting that impact of a variant on the individual depends on network context rather than simply the variant's effect on the gene. Analysis of network impact of genetic variants is a topic of recent interest, which this manuscript presents an interesting and novel take on. However, some important questions remain to be addressed.

Major comments:

The benefits of performing centrality and other analysis at the module level as opposed to gene level have yet not been demonstrated sufficiently. The "closeness centrality" analysis could be performed for gene nodes rather than modules, and may reveal the same results in a more straightforward and interpretable fashion. It would be useful to see simply the correlation between closeness centrality of each gene and frequency of LoF mutation. Even if modules reveal an interesting and distinct pattern from genes, it is worth comparing with the gene version.

Additional comments on module results: "Contrarily, deleterious variants that affect genes in peripheral modules of the network have a high likelihood of both cause minor distortions to the interactome and affect only non-essential functionalities". The authors did not provide analysis to distinguish between central nodes in central modules and peripheral nodes in central modules. Do mutations to genes that are not central themselves, but are members of central modules still follow the described pattern, or is it restricted to central genes? This reinforces my first comment.

Do these results generalize to networks other than the interactome tested here? There are many ways to construct a gene network, and it would not be a major undertaking to repeat the analysis using more network derivations, such as co-expression network, regulatory network, metabolic network, or co-functionality network. Even if the results do not extend to all tested network types, it would be very enlightening to compare the conclusions in at least one or two other network types. Regulatory variants, for instance, have been shown to have non-random distribution in both PPI and regulatory networks; do LoF variants similarly demonstrate these patterns regardless of network type?

Regarding the significance of number of interactions for genes with deleterious homozygous mutations compared to het or unaffected genes: if a rank-sum (Mann Whitney U) is used instead of Kruskal-Wallis, do the results remain significant?

For the claim "When the frequencies of individuals carrying a deleterious variant are compared, a trend of reduction in the number of connections with respect to the number of carrier individuals is observed (Figure 2, right)", the authors should provide a p-value. Is this a significant relationship?

One major concern with the claims regarding impact on network structure is the nature of the simulations performed. The authors state "we simulated for each population 1000 individuals with the same number of proteins carrying deleterious variants distributed randomly across the interactome for each population." This captures only the effect of number of mutations, and thus confounds the network effects of 1) the actual individual genes observed with deleterious variants in a population with 2) the co-occurrence of pairs or sets of variants in the population. For example, it is a very different scenario if depletion of network-altering mutation profiles is entirely due to fewer mutations found in individual central genes, or due to depletion of co-occurring mutations that would together alter the network more significantly (such as mutating two genes simultaneously that are redundant to each other, but essential to have at least one). The authors should run an additional simulation in which individuals are generated such that a mutation in gene A follows the true

probability of mutation in A found in the real population, but is drawn independently of all other mutations rather than following the real co-occurrence distribution.

It would be illuminating to investigate whether the patterns observed are consistent with signatures of selection. Do gene-level metrics of selection (conservation, dn/ds etc) also correlate with the specific network properties investigated such as centrality, and does selection fully explain the results observed here? If not, what else could explain them?

Minor comments:

In the intro, it is not yet clear how "less connections and lower centrality" is different from "peripheral", since number of connections is one standard metric of centrality (not the one the authors use, but this is not described until much later).

Figure 2c seems to collapse a lot of information in the 44-1290 bucket. can this be broken down further into more quantiles? Would a box or scatter plot be more revealing?

At the beginning of Results, briefly summarize what data defines "the interactome".

The intro doesn't lay out existing evidence/citations for the statement "However, the mechanisms by which specific deleterious variants can have a clear pathological effect when affect to some genes while in other ones are apparently innocuous, remain largely unknown." Are the LoF variants already known to be deleterious in some genes but not others, regardless of variant effect on the gene? Please provide citations to support this statement.

"A study of double synthetic lethal in yeast revealed that, while removal of two individual nodes have no effect in the yeast metabolic network, its combined removal produced a severe disruption in the information flow (Segre et al, 2005)." This statement is unclear - it sounds like the authors are claiming that *any* two individual nodes can be removed without disruption, which is not true. The cited study demonstrated this is true for some, not all, genes. If the authors wish to cover synthetic lethality literature, there are many large studies potentially worth citing (eg Costanzo 2010).

"This impact is inferred by measuring the changes in several fundamental network properties such as the loss of connections, the separation into isolated components and the increase of shortest path lengths. The values obtained for these parameters represent, in theory, normal, healthy interactomes." - have any previous studies shown that these values do in fact predict any property of fitness or health? Add citations if so, otherwise provide some additional justification in text.

The CLL analysis somewhat interesting but a bit distracting from the key results. It's moderately comforting that the conclusions from germline mutations do contrast with the CLL somatic mutations, but it's not a key analysis. These results could potentially be restricted to their own section, rather than mentioning them scattered throughout the manuscript.

In the discussion: "Conversely, damaged internal proteins cause more drastic alterations in the interactome topological properties and are most commonly associated to disease." - was association to disease discussed in this study except in relation to CLL? It would be interesting if the claim could be extended to known Mendelian disease mutations or other database of disease variants.

The manuscript needs to be more thoroughly proof-read for grammatical errors, I have listed a few below but there are several more:

Typo in the abstract: ".effect when occur." i think should be "effect when they occur"

Typo in intro: "Of importance particular"

"less connections" should be "fewer connections"

"independent evidences" should be "independent evidence"

Additional relevant citations:

Keinan, Alon, and Andrew G. Clark. "Recent explosive human population growth has resulted in an excess of rare genetic variants." *science* 336.6082 (2012): 740-743.

Battle, Alexis, et al. "Characterizing the genetic basis of transcriptome diversity through RNA-sequencing of 922 individuals." *Genome research* 24.1 (2014): 14-24.

Fraser HB, et al. 2002. Evolutionary rate in the protein interaction network. *Science* 296(5568): 750-752.

Jordan IK, et al. 2004. Conservation and coevolution in the scale-free human gene coexpression network. *Mol Biol Evol* 21(11): 2058-2070.

Gerstein MB, et al. 2012. Architecture of the human regulatory network derived from ENCODE data. *Nature* 489(7414): 91-100.

1st Revision - authors' response

16 June 2014

Point by point responses to referees

Reviewer #1:

The manuscript presents a carefully elucidated analysis of the effects of genetic variants on protein-protein interactome, through comparing sequence variant data from different populations. This computational analysis attempts to interpret why apparently deleterious variants are carried along in normal, healthy individuals. The sequencing data is used to compare natural variants to variants within chronic lymphocytic leukemia (CLL) patients (somatic and germline mutations). The interactome network they use is derived from literature data taken from curated interaction databases. The major conclusion is that the deleteriousness of a variant depends on the location of the protein in the interactome network. While the question addressed is of wide-reaching interest, many statements are not precisely made, and additional controls or analyses would be required to make the conclusion sound. The text would also need major editing and correction to convey the message optimally.

COMMENT

Major points:

- In the Introduction it is stated that disease proteins tend to be more central in interactome networks. This is actually in contradiction to the papers cited. Since the first demonstration in 2007 (Goh et al., PNAS) and many replications since, it is accepted that disease genes, except for cancer, locate on the periphery of the interactome. Cancer genes are the exception. This is a fundamental problem for this manuscript since the main generalization proposed made is that for all disease genes, cancer or non-cancer, the location of the affected protein in the interactome underlies the deleteriousness of the variant. "Disease proteins" mutated in cancer are more "central" compared to loss-of-function natural variants which are in proteins more on the periphery, but whether this conclusion holds for all disease genes needs to be clarified.

RESPONSE

Actually, the idea we wanted to convey is that mutations causing severe pathogenic effects (high mortality) tend to be more central in the interactome (an obvious case are mutations in essential genes). A large number of disease mutations do not have a strong pathogenic effect (alone or even in combination with other disease mutations) and the reason for this is, probably, that they do not affect essential functions located in internal regions of the interactome. This is the reason why in some cases disease mutations can also been found in peripheral regions of the interactome. We also observe a significant (although not so remarkable) relationship with the centrality in a particular

subset of diseases, composed by monogenic diseases with a more severe pathology, that we have included as suggested by both referees as well as to clarify the point as requested. We have rewritten the paragraph in the introduction and we have included an extra analysis of monogenic disease genes in results. As expected, gene products of monogenic diseases occupy an “intermediate” region of the interactome (not as internal as cancer genes, but clearly not peripheral like LoF genes in healthy individuals). Goh (2007) comment on disease genes refers to the unexpected non-centrality of many disease genes and their explanation was that they were not essential (while cancer genes are). Actually, non-essential genes present different degrees of potential deleteriousness, which can be correlated to their relative locations in the interactome and to their potential for disrupting the interactome when its function is lost. An evolutionary perspective, provided by Kim (2007, PNAS) argues along these lines. Kim et al. state: “Goh et al. (36) provide evidence that proteins involved in genetic diseases show little preference for either the center or the periphery. This is also consistent with our results. The diseases in their dataset that are of a genetic nature (e.g., leukemia, etc.)—i.e., they are ‘intrinsic’ diseases and are hence not involved with the environment—are also not that likely to be involved in adaptive evolution. Conversely, proteins that are involved in dealing with externally caused diseases (e.g., proteins involved in immune response) are likely to be on the cellular periphery.”

The title of the subsection where the analysis of monogenic diseases was included was changed to “Proteins affected by deleterious variants observed in normal population, in monogenic diseases and in cancer patients have different topological roles”. We have decomposed the former figure 3 into 2 figures (3 and 4). Current figure 3 now includes monogenic disease proteins and also proteins carrying somatic deleterious variants in CLL patients that have been classified as cancer drivers. We have included a supplementary table (2) with the genes of monogenic diseases and cancer drivers used in the study.

COMMENT

• As a test of the hypothesis that disease mutations are more central in the interactome network somatic mutations from CLL patients are employed. The problem here is that cancer proteins have been shown to differ greatly from non-cancer disease proteins with respect to network topology. To draw more general conclusions proteins associated with diseases other than cancer should also be analysed.

RESPONSE

We have included monogenic diseases in the analysis (see response to previous comment)

COMMENT

• The PPI networks employed in this work rely on literature-curated interactome (LCI) datasets. Judiciously a high-quality version of the interactome is used by keeping only PPIs supported by several methods. One criticism that has been made about LCI networks is that their topology is systematically biased because some proteins are more studied than others (Das & Yu, BMC Syst Biol 2012; Edwards et al, Nature 2011) It should be checked whether what is observed might be reflecting such bias. At the least discuss this point should be discussed.

RESPONSE

We completely agree with the referee in this point. However, for the sake of the reproducibility and the comparability to other studies we decided to use the most commonly used versions of the interactome (we obtained it from BioGRID, IntAct and MINT). We understand that by doing that we are accepting a certain degree of bias towards the most studied proteins, like many other papers published using this version of the interactome. In any case, the main conclusions of the work are about general properties of the network, and deal with location of affected proteins within the global structure of the interactome, therefore it is difficult to think that the general conclusions could be affected by biases caused by particular more studied proteins.

In any case, as the referee requested, we have added a paragraph in discussion mentioning this problem. We agree with the referee that it is worth mentioning it.

COMMENT

• Most of the work relies on the prediction of the deleterious effect of the mutations. The authors use stringent criteria to define such mutations. Assessing experimentally (or for instance using available 3D structures) a subset of these mutations would likely add to this paper.

RESPONSE

As recommended by the referee, we have included a subsection in methods “Validation of the threshold of deleteriousness”, where we validated the threshold used for declaring a variant as deleterious, and the corresponding subsection “In silico structural analysis of the impact of mutations in the proteins” in methods. A figure (Figure 1) and a table (Table 1) have been added as part of the subsection added in results.

COMMENT

o Please read the entire manuscript for syntax, grammar and tense mistakes. Some examples are provided here as minor points.
The legends and figures should more precisely indicate what is represented. For instance, is it the "number of mutations" or "average number of mutations per patient"?
Abstract, Last line, "system" should read "systems".
Introduction, Line 4, "Of importance particular" should be "Of particular importance".
Introduction, Paragraph 2, Line 25, "severe disruption" should be only under negative (but not positive) genetic interaction.
Results, second section, the title "Deleterious variants observed in normal population and cancer patients affect to proteins" should be "... affect proteins".
Same section, Paragraph 2, Line 3, "it is still" should be "is still".

RESPONSE

We have corrected the mistakes listed by the referee and rewritten the figure legends for a more precise indication of the content of the figures. We have also made an extensive grammatical correction of the text helped by a native English speaker.

COMMENT

Whenever possible, please provide the exact P values, but not just "p-value<0.01".

RESPONSE

Done. We have provided them in the text and in the figures.

Reviewer #2:

This manuscript presents a network-based analysis of loss-of-function variants in individual genomes from several populations. The authors provide evidence that the frequency of LoF variants correlate with network properties including connectivity and centrality, suggesting that impact of a variant on the individual depends on network context rather than simply the variant's effect on the gene. Analysis of network impact of genetic variants is a topic of recent interest, which this manuscript presents an interesting and novel take on. However, some important questions remain to be addressed.

Major comments:

COMMENT

The benefits of performing centrality and other analysis at the module level as opposed to gene level have yet not been demonstrated sufficiently. The "closeness centrality" analysis could be performed for gene nodes rather than modules, and may reveal the same results in a more straightforward and interpretable fashion. It would be useful to see simply the correlation between closeness centrality of each gene and frequency of LoF mutation. Even if modules reveal an interesting and distinct pattern from genes, it is worth comparing with the gene version.

RESPONSE

We preferred modules for two reasons: firstly for the sake of the visualization (the summarized representation of the complete interactome can easily be interpreted, while the representation of the complete interactome with all the proteins constitutes a nice picture but it is not informative), and secondly because we wanted to further connect the modules with functions. It is well known that these modules are sets of proteins highly connected among them and weakly connected to the rest of the interactome that tend to play specific functional roles. That is, they can be considered functional modules.

Having said this, we agree with the referee that the use of modules could include a potential bias that must be discarded. Therefore, we have followed the suggestion made and calculated the correlation (Spearman's rank correlation coefficient) between the three network properties (closeness centrality, the betweenness and the number of interactions) and the frequency of individuals affected. In order to take the same approach of over or under-representation of affected genes, instead of using modules, we have divided the interactome in (approximately) concentric spheres and calculated these values for each sector delimited between two spheres. We have added this result at the end of the subsection "Variants observed in normal individuals tend to occur at the periphery of the interactome". Also Supplementary Figure 2 documents the same pattern than Figure 7 (relationship of mutated gene modules with centrality), but without using modules.

COMMENT

Additional comments on module results: "Contrarily, deleterious variants that affect genes in peripheral modules of the network have a high likelihood of both cause minor distortions to the interactome and affect only non-essential functionalities". The authors did not provide analysis to distinguish between central nodes in central modules and peripheral nodes in central modules. Do mutations to genes that are not central themselves, but are members of central modules still follow the described pattern, or is it restricted to central genes? This reinforces my first comment.

RESPONSE

As mentioned in the previous comment, in order to better clarify the relationship between centrality and the frequency of observation of deleterious mutations we have also repeated the study without taking into account the modules. We have subdivided the interactome into four regions of different centrality (from more external to more central). We have added a comment in the corresponding section of results as well as an additional figure (Supplementary Figure 2)

COMMENT

Do these results generalize to networks other than the interactome tested here? There are many ways to construct a gene network, and it would not be a major undertaking to repeat the analysis using more network derivations, such as co-expression network, regulatory network, metabolic network, or co-functionality network. Even if the results do not extend to all tested network types, it would be very enlightening to compare the conclusions in at least one or two other network types. Regulatory variants, for instance, have been shown to have non-random distribution in both PPI and regulatory networks; do LoF variants similarly demonstrate these patterns regardless of network type?

RESPONSE

The main idea behind the manuscript was to take a systems biology approach to the problem of explaining the apparently high occurrence of deleterious mutations that cause loss of function in the affected proteins but cause no apparent phenotype to the healthy carrier. We use the interactome as a scaffold that defines functional modules (the neighborhoods). We have then studied the effect of the loss of function of the affected proteins within the context of the interactome and estimate the damage caused to its structure. As we state in the introduction: "...our hypothesis is that the actual interactome topology could be buffering the impact of deleterious variants, allowing thus what it seems to be a high mutation load."

This study was specifically oriented to understand the role of the interactome in buffering the impact of deleterious mutation, and we have used network properties of the interactome to measure the impact. But we do not think that the conclusions achieved on the network properties in the interactome can be directly transferred to other networks that represent completely different biological backgrounds.

In addition, the comparison could be complex in some cases. For example, co-expression networks are tissue-dependent and in some cases are also disease-dependent (e.g., cancer). Regulatory networks and metabolic networks are directed networks, opposite to the interactome or the co-expression network, and account for completely different biological processes (although in some cases there exist some correlations).

Therefore, we think that the inclusion of other networks falls outside the scope of the manuscript.

COMMENT

Regarding the significance of number of interactions for genes with deleterious homozygous mutations compared to het or unaffected genes: if a rank-sum (Mann Whitney U) is used instead of Kruskal-Wallis, do the results remain significant?

RESPONSE

Yes, they are still significant. Actually we have changed the test and we apply now the U Mann-Whitney test with FDR correction for multiple testing, which seems more appropriate to compare pairs of distributions. Figure 3 includes such comparisons.

COMMENT

For the claim "When the frequencies of individuals carrying a deleterious variant are compared, a trend of reduction in the number of connections with respect to the number of carrier individuals is observed (Figure 2, right)", the authors should provide a p-value. Is this a significant relationship?

RESPONSE

We calculated a Spearman tests which confirmed that the trend was low but significant. A paragraph has been added with the results of the test in the subsection "Proteins affected by deleterious variants observed in normal population, in monogenic diseases and in cancer patients have different topological roles". The former Figure 2c is now Figure 4.

COMMENT

One major concern with the claims regarding impact on network structure is the nature of the simulations performed. The authors state "we simulated for each population 1000 individuals with the same number of proteins carrying deleterious variants distributed randomly across the interactome for each population." This captures only the effect of number of mutations, and thus confounds the network effects of 1) the actual individual genes observed with deleterious variants in a population with 2) the co-occurrence of pairs or sets of variants in the population. For example, it is a very different scenario if depletion of network-altering mutation profiles is entirely due to fewer mutations found in individual central genes, or due to depletion of co-occurring mutations that would together alter the network more significantly (such as mutating two genes simultaneously that are redundant to each other, but essential to have at least one). The authors should run an additional simulation in which individuals are generated such that a mutation in gene A follows the true probability of mutation in A found in the real population, but is drawn independently of all other mutations rather than following the real co-occurrence distribution.

RESPONSE

This is a very good point. Actually, before submitting the manuscript, we were discussing whether to include this simulation or not. Finally, for the sake of the simplicity we decided not to include it. But the referee is right in observing that the combined effect of the mutations should be tested as well. Therefore, we have included it in the corrected version. Now we have a second scenario of simulation in which we generate simulated individuals with deleterious mutations only in proteins already observed in the healthy individuals but randomly distributed (at the frequencies of occurrence in the population). In that way we remove the actual combinations but we do not introduce mutations in proteins which never appear mutated in healthy individuals. We added the corresponding paragraph in the subsection "Effect of deleterious variants observed in normal individuals over the interactome structure", we have produced a new version of the figure (now Figure 5) and we have added the description of the new simulation to the "Assessing the effect caused by homozygote deleterious variants in on the interactome structure" subsection of Methods

COMMENT

 It would be illuminating to investigate whether the patterns observed are consistent with signatures of selection. Do gene-level metrics of selection (conservation, dn/ds etc) also correlate with the specific network properties investigated such as centrality, and does selection fully explain the results observed here? If not, what else could explain them?

RESPONSE

 Following the suggestion of the referee, we have studied the distribution of proteins under positive ($dN/dS > 1$) and negative ($dN/dS < 1$) selective pressures along the three main network properties used in this study, confirming at the level of these properties the trend previously observed by Kim et al., 2007 of occurrence of positive selection in the periphery on the interactome and negative selection in its internal part. We have added a supplementary figure (suppl. Fig 1) that depicts this trend. Obviously, there is a connection on one hand between selection and phenotype and, on the other hand, between selection and position in the network. Selection have fewer constrains to operate over the less essential proteins that are more peripheral.

COMMENT

 In the intro, it is not yet clear how "less connections and lower centrality" is different from "peripheral", since number of connections is one standard metric of centrality (not the one the authors use, but this is not described until much later).

RESPONSE

 We wanted to express that connectivity (regardless of the location of the protein in the interactome) was known to be related with pathologic effect. Beyond this, we observed that the location in the interactome is important by itself. Obviously both properties are not independent. We have removed "and lower centrality" for the sake of clarity.

COMMENT

 Figure 2c seems to collapse a lot of information in the 44-1290 bucket. can this be broken down further into more quantiles? Would a box or scatter plot be more revealing?

RESPONSE

 The referee is right: the last point seems a bit out of scale at a resolution corresponding to four quartiles. We tried with more bins but, unfortunately, the low numbers of mutations observed simultaneously in many individuals made the plot a bit erratic and, although the trend was significant (see four comments above), the visual representation was confusing. Finally, we have broken the last bin down into two equal parts to avoid an arbitrary partition. We hope that the figure is more readable now. We have changed the figure legend accordingly, which now is Figure 4, and contains not only connectivity but also betweenness and closeness centrality.

COMMENT

 At the beginning of Results, briefly summarize what data defines "the interactome".

RESPONSE

 OK, done. Instead of including this summary at the beginning of Results, we have included it in the second subsection "Deleterious variants observed in the populations", where the interactome is mentioned for the first time.

COMMENT

 The intro doesn't lay out existing evidence/citations for the statement "However, the mechanisms by which specific deleterious variants can have a clear pathological effect when affect to some genes while in other ones are apparently innocuous, remain largely unknown." Are the LoF variants already known to be deleterious in some genes but not others, regardless of variant effect on the gene? Please provide citations to support this statement.

RESPONSE

 What we mean is that the previous cited references (Nothnagel et al, 2011; Xue et al, 2012) describe an overabundance of LoF variants but do not explain why they exist without causing pathologies while other variants with similar deleterious effect on the proteins have severe pathological consequences. We have quoted the references at the end of the sentence.

COMMENT

 "A study of double synthetic lethal in yeast revealed that, while removal of two individual nodes have no effect in the yeast metabolic network, its combined removal produced a severe disruption in the information flow (Segre et al, 2005)." This statement is unclear - it sounds like the authors are claiming that *any* two individual nodes can be removed without disruption, which is not true. The cited study demonstrated this is true for some, not all, genes. If the authors wish to cover synthetic lethality literature, there are many large studies potentially worth citing (eg Costanzo 2010).

RESPONSE

 The referee is right. We have rephrased the sentence to the meaning suggested by the referee and also added the suggested reference.

COMMENT

 "This impact is inferred by measuring the changes in several fundamental network properties such as the loss of connections, the separation into isolated components and the increase of shortest path lengths. The values obtained for these parameters represent, in theory, normal, healthy interactomes." - have any previous studies shown that these values do in fact predict any property of fitness or health? Add citations if so, otherwise provide some additional justification in text.

RESPONSE

 Our apologies, because clearly we did not conveyed the desired meaning in the sentence. It is not that the particular network values predict a healthy status. The healthy status is a property of the dataset observed (1000 genomes are supposed to be healthy individuals, or CLL are diseased individuals), and we measured the network properties in these individuals to look for a relationship between healthy/diseased status and network properties. We have rephrased the last sentence of the paragraph to: "The values obtained for these parameters in the 1000 genomes and MGP1 populations correspond, in theory, to normal, healthy interactomes."

COMMENT

 The CLL analysis somewhat interesting but a bit distracting from the key results. It's moderately comforting that the conclusions from germline mutations do contrast with the CLL somatic mutations, but it's not a key analysis. These results could potentially be restricted to their own section, rather than mentioning them scattered throughout the manuscript.

RESPONSE

 We have followed the recommendation of the referee and we removed the comments on somatic CLL mutations from other parts of the text.

COMMENT

 In the discussion: "Conversely, damaged internal proteins cause more drastic alterations in the interactome topological properties and are most commonly associated to disease." - was association to disease discussed in this study except in relation to CLL? It would be interesting if the claim could be extended to known Mendelian disease mutations or other database of disease variants.

RESPONSE

 Thanks for raising this point. As per request of the first referee too, we have added monogenic diseases to the study. The results obtained are similar to the observed for CLL, although the effect is

a bit less drastic. We have rewritten the paragraph in the introduction and we have included an extra analysis of monogenic disease genes in results.

COMMENT

The manuscript needs to be more thoroughly proof-read for grammatical errors, I have listed a few below but there are several more:

Typo in the abstract: ".effect when occur." i think should be "effect when they occur"

Typo in intro: "Of importance particular"

"less connections" should be "fewer connections"

"independent evidences" should be "independent evidence"

RESPONSE

As also requested by the first referee, we have made an extensive grammatical correction of the text helped by a native English speaker.

2nd Editorial Decision

01 August 2014

Thank you again for submitting your work to Molecular Systems Biology. We have now heard back from the two referees who agreed to evaluate your manuscript. As you will see from the reports below, the main concerns of both reviewers have been satisfactorily addressed. However, they still list a few relatively minor issues that we would ask you to address in a revision of the manuscript. In particular, reviewer #1 mentions that the discussion should clearly explain how well-studied and under-studied proteins could affect the topological properties of the presented network. Moreover, both reviewers refer to the need to improve text quality.

REFeree REPORTS

Reviewer #1:

The manuscript presents carefully elucidated and well-controlled analyses of genetic variant effect on protein-protein interactome, proposing that network topological properties partly explain the subsistence of deleterious variation in human populations. This revised manuscript addresses the main concerns raised at the time of the initial review. In this revised version, the authors have added new analyses and notably two important controls. First, they have added support to the prediction of deleteriousness of lof mutations based on 3D structures. Second, they compare the network topology of products of non-disease genes with lof mutations and of cancer genes to those of genes associated with monogenic diseases. They have also modified and clarified the text and expanded the discussion.

There are still two minor points I am concerned with and that the authors need to take care of to further improve their manuscript and make it acceptable for publication:

- As mentioned previously, since the authors use interactome networks based on literature-curation, their topology is likely affected by inspection biases inherent to literature datasets. The authors have modified the discussion to reflect this but do not do it appropriately. They argue that the network they study might be biased towards well-studied proteins but that this should not affect the topological properties of proteins in that network. This is not correct. The "true" degree of a protein will most likely be more underestimated for understudied proteins than for well-studied proteins. In other words, the observed degree of a node in this network results from the combination of two parameters, first its true degree in the full interactome, and second, the level to which the corresponding gene/protein has been investigated in the literature (and curated). The combination of these two effects might for instance very well explain why the difference is so large when comparing cancer genes to other classes. Since the degree of a node has a crucial impact on every topological property, this should most likely bias the results reported in this study and need to be at least mentioned correctly in the discussion.

- Despite some improvements, the text remains of poor quality. There are still too many grammar and spelling mistakes, some calls to the figure panels are wrong, a few sentences do not make sense. This should be fixed much more carefully so that the quality of the text would reach the quality and interest of the reported analyses.

Reviewer #2:

The authors have substantially revised the manuscript substantially, addressing the majority of the concerns previously stated, including additional simulations and evaluation of monogenic traits.

One suggestion:

Supplementary Figure 1:

Within genes where $dn/ds < 1$, does the strength of selective constraint correspond to the reported network properties? This would be a more standard analysis rather than comparing $dn/ds > 1$ to $dn/ds < 1$. An additional subfigure should be added grouping genes into buckets by (negative) selective constraint for $dn/ds < 1$.

There remain a few typos and grammatical errors in the text, so a final pass of proof-reading is needed. Errors I noticed:

- * "observation affects to proteins or the clustering" unclear due to grammar
- * a few places the authors refer to "U Mann Whitney", should be "Mann Whitney U"

2nd Revision - authors' response

23 August 2014

Point by point responses to the reviewers

Reviewer #1:

The manuscript presents carefully elucidated and well-controlled analyses of genetic variant effect on protein-protein interactome, proposing that network topological properties partly explain the subsistence of deleterious variation in human populations. This revised manuscript addresses the main concerns raised at the time of the initial review. In this revised version, the authors have added new analyses and notably two important controls. First, they have added support to the prediction of deleteriousness of lof mutations based on 3D structures. Second, they compare the network topology of products of non-disease genes with lof mutations and of cancer genes to those of genes associated with monogenic diseases. They have also modified and clarified the text and expanded the discussion.

There are still two minor points I am concerned with and that the authors need to take care of to further improve their manuscript and make it acceptable for publication:

COMMENT

 - As mentioned previously, since the authors use interactome networks based on literature-curation, their topology is likely affected by inspection biases inherent to literature datasets. The authors have modified the discussion to reflect this but do not do it appropriately. They argue that the network they study might be biased towards well-studied proteins but that this should not affect the topological properties of proteins in that network. This is not correct. The "true" degree of a protein will most likely be more underestimated for understudied proteins than for well-studied proteins. In other words, the observed degree of a node in this network results from the combination of two parameters, first its true degree in the full interactome, and second, the level to which the corresponding gene/protein has been investigated in the literature (and curated). The combination of these two effects might for instance very well explain why the difference is so large when comparing cancer genes to other classes. Since the degree of a node has a crucial impact on every topological property, this should most likely bias the results reported in this study and need to be at least mentioned correctly in the discussion.

RESPONSE

 As requested we have modified the corresponding paragraph in the discussion and included this new paragraph: "Despite being carefully curated, the use of the interactome always entails a risk of obtaining results biased towards well-studied biological processes (Das & Yu, 2012; Edwards et al, 2011). Thus, the true degree of understudied proteins could be underestimated with respect to that of the well-studied proteins. It might be argued that this effect could, for example, inflate the differences in network parameters between cancer genes and other classes. Nevertheless, it is expected that the interactome used here suffer to a lesser extent from this bias given that protein interaction data were only retrieved from raw data repositories, avoiding knowledge-based sources." We hope that the text is now properly conveying the idea suggested by the referee.

COMMENT

 - Despite some improvements, the text remains of poor quality. There are still too many grammar and spelling mistakes, some calls to the figure panels are wrong, a few sentences do not make sense. This should be fixed much more carefully so that the quality of the text would reach the quality and interest of the reported analyses.

RESPONSE

 We apologize again for this. We have now extensively revised the text quality. A native English speaker expert in scientific paper correction has helped us. We hope that the text quality of the present version of the manuscript is now acceptable.

Reviewer #2:

The authors have substantially revised the manuscript substantially, addressing the majority of the concerns previously stated, including additional simulations and evaluation of monogenic traits.

One suggestion:

COMMENT

 Supplementary Figure 1:

Within genes where $dn/ds < 1$, does the strength of selective constraint correspond to the reported network properties? This would be a more standard analysis rather than comparing $dn/ds > 1$ to $dn/ds < 1$. An additional subfigure should be added grouping genes into buckets by (negative) selective constraint for $dn/ds < 1$.

RESPONSE

 We agree with the referee that this representation better conveys the idea that selective pressure is related to location within the interactome. After several tests we concluded that three buckets (corresponding to positive, neutral and negative selection) is the optimal representation. The reason for this can be found in the new Figure E1C: the distribution of genes along the dN/dS axis is not is not continuous but rather follows a trimodal distribution.

COMMENT

 There remain a few typos and grammatical errors in the text, so a final pass of proof-reading is needed. Errors I noticed:

* "observation affects to proteins or the clustering" unclear due to grammar

* a few places the authors refer to "U Mann Whitney", should be "Mann Whitney U"

RESPONSE

 We have tried to correct all the grammatical errors and improve the quality of the text with the help of a native English speaker expert in scientific paper correction.